# Evolution of an Alpine proglacial river during seven decades of deglaciation

Livia Piermattei[1,2,3], Tobias Heckmann[3], Sarah Betz-Nutz[3], Moritz Altmann[3], Jakob Rom[3], Fabian Fleischer[3], Manuel Stark[3], Florian Haas[3], Camillo Ressl[4], Michael H. Wimmer[4], Norbert Pfeifer[4], Michael Becht[3]

[1]Swiss Federal Institute for Forest, Snow and Landscape Research (WSL), Birmensdorf, 8903, Switzerland
[2]Department of Geosciences, University of Oslo, Oslo, 0371, Norway
[3]Physical Geography, University of Eichstätt-Ingolstadt, Eichstätt, 85072, Germany
[4]Department of Geodesy and Geoinformation, TU Wien, Wien, 1040, Austria

*Correspondence to*: Livia Piermattei (livia.piermattei@wsl.ch)

**Abstract.** Alpine rivers have experienced considerable changes in channel morphology over the last century. Natural factors and human disturbance are the main drivers of changes in channel morphology that modify natural sediment and flow regimes at local, catchment, and regional scales. In glaciated catchments, river sediment loads are likely to increase due to increasing snow and glacier melt runoff, facilitated by climate changes. Additionally, channel erosion and depositional dynamics and patterns are influenced by sediment delivery from hillslopes, and sediment in the forefields of retreating glaciers. In order to reliably assess the magnitudes of the channel-changing processes and their frequencies due to recent climate change, the investigation period needs to be extended to the last century, ideally back to the end of the Little Ice Age. Moreover, a high temporal resolution is required to account for the history of changes in channel morphology and for better detection and interpretation of related processes. The increasing availability of digitised historical aerial images and advancements in digital photogrammetry provides the basis for reconstructing and assessing the long-term evolution of the surface, in terms of both planimetric mapping and the generation of historical digital elevation models (DEMs).

The main issue of current studies is the lack of information over a longer period. Therefore, this study contributes to research on fluvial sediment changes by estimating the sediment balance of a main Alpine river (Fagge River) in a glaciated catchment (Kaunertal, Austria) over nineteen survey periods from 1953 to 2019. Exploiting the potential of historical multi-temporal DEMs combined with recent topographic data, we quantify 66 years of geomorphic change within the active floodplain, including erosion, deposition, and the amounts of mobilised sediment. Our study focuses on a proglacial river that is undergoing a transition phase, resulting from an extensive glacier retreat of approximately 1.8 kilometres. This has led to the formation of new channel networks, and an overall negative cumulative sediment balance for the entire study area. We found that high-magnitude meteorological and hydrological events associated with local glacier retreats have a significant impact on the sediment balance. The gauge record indicates an increase in such events, as well as in runoff and probably in sediment transport capacity. Despite this, the sediment supply has declined in the last decade, which can be attributed to a lower

contribution of the lateral moraines coupled to the channel network, and less sediment sourced from the melting Gepatsch Glacier as evidenced by roches moutonnées exposed in the current/most recent forefield. Nonetheless, we observed significant erosion in the tributary, leading to the transport of sediment downstream. Overall, this study enhances our understanding of the complexity of sediment dynamics in proglacial rivers across various spatial and temporal scales and their relationship to climate change factors.

## 1 Introduction

Alpine rivers play an essential role in Alpine sediment cascades; as "conveyor belts", they transfer sediments from their catchment towards their outlets. Their morphology, and changes thereof through erosion and deposition, are mainly controlled by river discharge, sediment supply, and valley morphology. Key discharge characteristics, such as the frequency and magnitude of peak flows, including extreme events, are governed by hydro-meteorological forcing, contributions of glacier and snow melt (Antoniazza et al., 2022), and catchment characteristics (topography, landcover; e.g. Mao et al., 2009; Comiti et al., 2011). Sediment supply to a river reach can be derived through different geomorphic processes from various sources and deposits, including a glacier, the catchment hillslopes, and the floodplain itself (e.g. Beylich & Laute, 2015). Hillslope-channel and within-channel coupling determine the delivery of these sediments to any specific river reach (Cavalli et al., 2013; Heckmann & Schwanghart, 2013); this is influenced by the marked downstream gradients of hydrological and geomorphological processes (Gurnell et al., 1999). In proglacial rivers, defined as river reaches within the limits of the Little Ice Age glaciated area (Carrivick et al., 2018), glacier geometry and dynamics, and sediment supply from the glacier terminus strongly impact river morphodynamics (Ashworth & Ferguson, 1986; Orwin & Smart, 2004); Marren & Toomath (2014), however, emphasise the dominating role of topographic forcing over changes in hydraulic and sediment supply conditions in the short term.

Over the last century, Alpine rivers have undergone remarkable changes due to natural factors and anthropogenic disturbances (Liébault & Piégay, 2002; Comiti et al., 2011; Marchese et al., 2017; Llena et al., 2020). Among natural factors, the main driver of changes in channel morphology, sediment availability, and discharge characteristics is the ongoing reduction of glacier volume facilitated by climate change. The glacial retreat leads to the emergence and growth of proglacial margin morphologies at the interface between the glacier and the river system (Heckmann et al., 2016; Carrivick & Heckmann, 2017). Consequently, this has implications for sediment delivery from deglaciated hillslopes (e.g. lateral moraines) and rock walls, which influences channel erosion and depositional dynamics and patterns. Other natural control factors especially related to climate change potentially causing morphological changes in Alpine rivers are increasing temperatures, altered precipitation patterns and regime controlling flood and drought (frequency, magnitude, timing, and altitude), and changes in snow cover and seasonal snow melting (Beniston 2006; Hock et al., 2019). Climate change and human activities also induce changes in land cover that may facilitate or impede runoff generation on the one hand, and the erosion and transfer of sediments on the other (Liébault and Piégay, 2002; Starkel, 2002). Sediment connectivity, i.e. the degree of coupling between upstream/upslope

sediment sources with a given point within the channel network, governs the delivery of sediments towards the outlet, and the propagation of changes through the catchment ("transmission sensitivity", Fryirs 2017), which is important within the context of ongoing climate, land cover, and geomorphic changes. Knight and Harrison (2018, p. 1992) highlight a lack of "interconnections between the elements of glaciers, meltwater, and sediments in studies of deglacierising mountains".

The reconstruction of changes that occurred in past decades and their interpretation in light of potential drivers could help to
fill this knowledge gap and to improve our understanding of the linkages between channel geomorphic adjustment, drivers, and the different sediment sources. This task is challenging due to the complexity of the mountain environment, the high dynamics nature of the fluvial system, and the wide range of scales on which geomorphological processes occur (e.g. Lane et al., 2017). Nonetheless, quantifying the past and present impact of natural factors and human disturbance on Alpine rivers is necessary to understand and predict how sediment production, storage, and transfer from sediment sources to the catchment
outlet might change in the future. For example, future scenarios include water scarcity and, as a consequence, a decrease in sediment transport capacity, as the maximum runoff from glacier long-term storage ("peak water") has already been or will be reached in the coming decades (Huss and Hock, 2018). No less importantly, the sediment supply in the catchment can represent a source of risk during flood events (Rickenmann & Koschni, 2010). In order to reliably assess the magnitudes of the channel changes processes and/or their frequencies due to recent climate change, the investigation period needs to be extended as much
as possible. In addition, a high temporal resolution is required to account for the history of changes in channel morphology and for better detection and interpretation of related processes.

Multi-temporal digital elevation models (DEMs) are widely used to calculate the elevation change over time and thus to investigate the spatial distribution of positive and negative sediment change and interpret these changes in terms of sediment transfer (e.g. Leyland et al., 2017; Vericat et al., 2017). Remote sensing technologies such as airborne and terrestrial laser
scanning (ALS, TLS) as well as aerial images provide high spatial and temporal resolution DEMs. For the quantitative analysis of past morphodynamics, the increasing availability of digitised historical aerial images, together with advancements in digital photogrammetry such as Structure from Motion (SfM), provides the basis for generating historical DEMs depicting the historical state of the fluvial system (Bakker and Lane, 2017). Schiefer and Gilbert (2007) were among the first to use digital photogrammetric techniques on historical aerial photographs for the reconstruction of past surface elevations and the
quantification of morphometric landscape changes in proglacial settings and glacial mass balances. Their dataset includes DEMs from twelve historical aerial photographs spanning the 50-year period from 1947 to 1997. However, their analysis focuses on the glacier and on hillslopes rather than on the floodplain of the proglacial river. Micheletti et al. (2015) produced photogrammetric DEMs from aerial imagery acquired between 1967 and 2012 (eight datasets). They used the resulting DEMs of difference to quantify surface changes and interpreted the latter with respect to cooling and warming episodes with the help
of a geomorphological map. The most conspicuous changes affected glaciers and periglacial landforms, whereas changes in the fluvial system were less clear, which the authors attributed to the smaller magnitude of those changes. Recently, Anderson and Shean (2021) used historical aerial images combined with LiDAR and satellite data to compare the geomorphological change in four deglaciating catchments at approximately decadal intervals.

In this study, we analyse past and recent changes in the morphology and the sediment storage of the main proglacial channel network of the Kaunertal catchment, Austria, with the overall aim to identify links between channel changes, sediment availability/delivery, and hydro-meteorological forcing. We hypothesise that river sediment loads are likely to have increased due to increasing snow and glacier melt runoff. In addition, we investigate sediment delivery from adjacent steep lateral moraines and a tributary channel that are hypothesised to contribute significantly to sediment supply to the proglacial channel network. To test these hypotheses, we reconstruct and quantify 66 years of sediment and river changes in the glacier forefield between 1953 and 2019 by applying the morphological method (Vericat et al., 2017) to DEMs from historical and digital aerial images and LiDAR that span inter-survey periods ranging from one month to 16 years (Fig. 1). For each period, we delineated the active floodplain, the lateral moraines, and the glacier extent and quantified the sediment changes in terms of erosion, deposition, and net balance. We do not know the amount of sediment (and changes thereof) supplied by the melting glacier, hence we can only quantify net storage changes in the channel network between the glacier and the outlet of our study area. The contribution of the glacier, however, can be substantial (60 % of suspended load according to Leggat et al., 2015). In order to assess the hydrological forcing, we also analysed the discharge record of a gauging station, focusing on discharge peaks and strong events, seasonal variation, and total trend.

From a technical perspective, the study also aims to highlight the potential of using historical aerial images to generate very high-resolution DEMs (1 m) for accurate and precise estimation of channel changes and associated sediment load in an Alpine catchment.

## 2 Methods

### 2.1 Study area

For the purpose of our study, we focus on the glacier forefield of the Gepatsch Glacier, located in Kaunertal in the Oetztal Alps (Fig. 1). The Gepatsch Glacier is the second largest glacier in Austria, which has suffered a continuous retreat after the phase of advance around 1980, as is the case for the majority of the valley glaciers in the Alps (Zemp et al., 2008). The Gepatsch Glacier terminus constitutes the source of the Fagge River, the main draining river in the catchment, which ends on the reservoir lake built in the early 1960s. The Fagge River mainly passes through a landscape characterised by lateral moraines, till- and scree-covered hillslopes, and bedrock landforms. At the end of the LIA (~1855) (Gross, 1987; Haeberli et al., 2019), the Gepatsch glacier covered 38 % of the catchment (62 km$^2$), but only 30 % in 1953. The catchment covers an altitude from 1810 m, measured at the outlet of the Fagge into the reservoir, up to 3583 m at the Hochvernagtspitze. With respect to geology, the catchment is located in the crystalline basement of the Oetztal massif, which is composed of gneisses and granite (Schöber and Hofer, 2018). Due to the high elevation, vegetation in the catchment is scarce. The study area is characterised by a typical dry high-alpine climate (Groh and Blöthe, 2019). The area of interest indicated in Figure 1 covers an area of 4.6 km$^2$ and extends from the bridge built around 1980 up to a topographic high in the southeast, which separated

the glacier terminus into two parts already in 1920 (Fig. 1). The elevation within the area of interest ranges from 1965 m to 3043 m. The red line in Figure 1 shows the main channel of the Fagge River (as of August 2017).

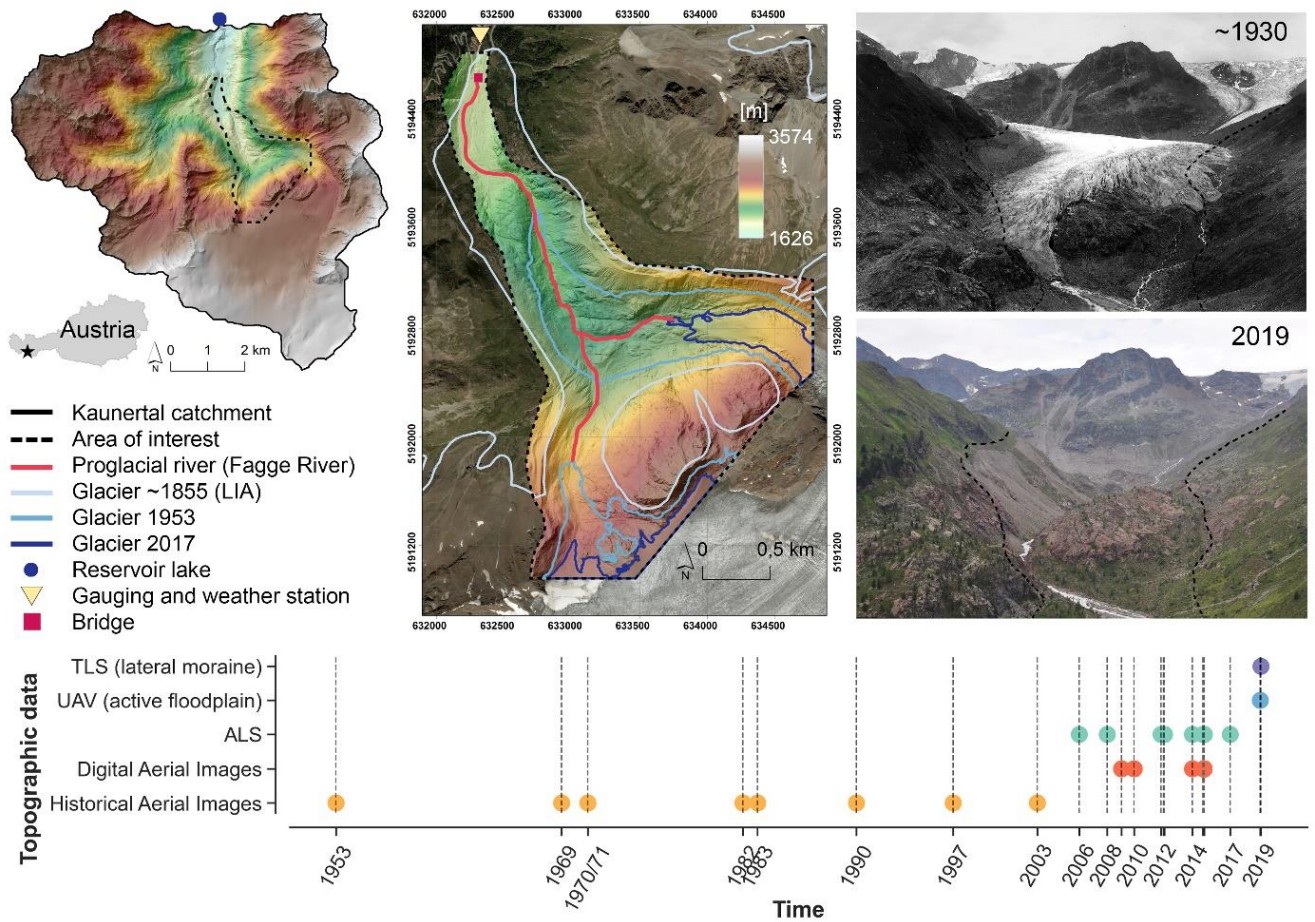

**Figure 1: Kaunertal catchment in Austria and the geographic setting of the area of interest (dashed black line) together with the Gepatsch Glacier extent at the end of LIA (1855), in 1953 and 2017, the active floodplain centreline (i.e. thalweg) as of 2017 with the**
**bridge location downstream, the location of the gauging and weather station and the reservoir lake. The elevation model refers to the year 2017 and the orthophoto is from 2015. On the right, two photos show a view of the proglacial area in 1930 and 2019 with the dashed line indicating the area of interest. Since 2003, the active floodplain has split into two branches as the glacier retreated. The graph at the bottom indicates the time step of each generated DEM and the corresponding topographic techniques. Groß and Patzelt (2015) is the source of the LIA glacier extent. Martin Frey, a local archivist of the Kaunertal, provided the historical photo,**
**but the photographer is unknown.**

## 2.2 Topographic data and generation of multi-temporal DEMs

Topographic data were obtained through different high-resolution topographic measurement techniques such as historical and digital aerial images, uncrewed aerial vehicle (UAV) images, and ALS and TLS. The acquisition date for each technique and the investigated time steps are illustrated in Fig. 1. In the following paragraphs, an overview of the data used is described

whereas a detailed description of the photogrammetric and LiDAR data characteristics, processing and post-processing workflow for each remote sensing technique is reported in the supplements (Sect. S1, Table S1, and Table S2).

Airborne and UAV images were processed to generate DEMs and orthophotos. The historical aerial images cover eight periods from 1953 to 2003: 1953, 1969, 1970/71, 1982, 1983, 1990, 1997, and 2003. These images are greyscale with the exception of the 2003 dataset, which contains RGB information. Airborne digital aerial images were collected in 2009 and 2010 with

RGB and NIR information. Additional available aerial images of 2014 and 2015 were processed only with the purpose of generating high-resolution orthophotos as DEMs over the same period were generated from ALS. In summer 2019, a UAV was deployed to acquire images of some segments of the main river. In parallel with the UAV flights, we surveyed the lateral hillslope adjacent to the braided river with a terrestrial laser scanner (Altmann et al., 2020).

ALS data were acquired in the years 2006, 2008, 2012, 2014, 2015, and 2017 in the summer months to avoid large-scale snow

cover. In 2012 and 2015, additional ALS data were collected at the end of the summer after a strong rainfall event. In total nine ALS DEMs were analysed. Because of its superior quality compared to the other ALS data (i.e. point cloud density and post-processing through strip adjustment), the ALS 2017 point cloud was used as a reference for i) picking the 3D coordinates of the ground control points (GCPs) used in the photogrammetric processing, and ii) computing the point cloud co-registration of photogrammetric and LiDAR point cloud.

From photogrammetric and LiDAR data we obtained high-density three-dimensional (3D) point clouds. To reduce the size of the files, only the 3D point clouds within the area of interest were post-processed for generating DEMs. The post-processing steps consisted of filtering vegetation and noise, point cloud decimation, and co-registration with the reference ALS 2017 data (see S1). The co-registration of the point clouds was performed through an iterative closest point (ICP) algorithm in stable areas distributed all around the proglacial area. All the co-registered point clouds were interpolated into a raster using a robust

moving window with 1 m resolution for the historical aerial images (i.e. DEMs from 1953 to 2003) and 0.5 m for the remaining data (from 2006 to 2017). Gap filling was applied to all the DEMs. The mentioned 3D processing and analysis were done with Opals (Pfeifer et al., 2014). All the topographic data are in the European Terrestrial Reference System 1989 (ETRS89) / Universal Transverse Mercator zone 32 north coordinate reference system (EPSG-code: 25832) in ellipsoid height (i.e. metres above ellipsoid).

**2.3 Mapping of glacier extent, active floodplain, and lateral hillslope**

For the quantification of erosion and deposition specifically in the river, and in order to separately quantify sediment contributions from the adjacent hillslopes, we mapped the active floodplain and the selected hillslope sections. This was necessary for every time step between 1953 and 2019, as the boundary between the active floodplain and depositional landforms at the foot of the adjacent hillslopes was seen to shift due to increased deposition from the hillslope on one hand

and undercutting by the river on the other. Sediment sources to the Fagge River system are the glacier and connected lateral hillslopes. An illustration of these landforms together with the active floodplain is shown in Fig. 2.

Our object of interest is the active floodplain, which includes the river channel, the floodplain with the lateral riverbank slopes, the mid and lateral bars that are un-vegetated or sparsely vegetated, and the vegetated island in the braided system downstream (Fig. 2b). Note that the vegetation was filtered out for the analyses.

The active floodplain is divided into six river reaches (Fig. 2a) defined as relatively homogeneous river sections along which current boundary conditions are sufficiently uniform (Brierley and Fryirs, 2005). Table 1 displays the main characteristics of these reaches as of 2017. The mean slope of the active floodplain is derived from the generalised centreline of the active floodplain (Fig. 1). In addition, we provide the mean slope of the mapped floodplain, characterised by the presence of steep riverbanks and terrace edges. The active floodplain polygon extends from the bridge built downstream at the beginning of

1980 (reach 1, Fig. 2b) to the eastern glacier terminus as of 2017 (reach 4, Fig. 2f) and to the southern glacier terminus as of 1953 (reach 6, Fig. 2e). The mapping of the southern active floodplain polygons does not change with the south glacier terminus fluctuations because snow patches often covered the channel. The mapping was performed based on various datasets, including a set of automatically generated breaklines of the slope raster, the DEM hillshade, orthophotos, and the intensity raster of the LiDAR data. Furthermore, the water surface within the active floodplain was mapped using orthophoto and LiDAR intensity

values for selected time steps.

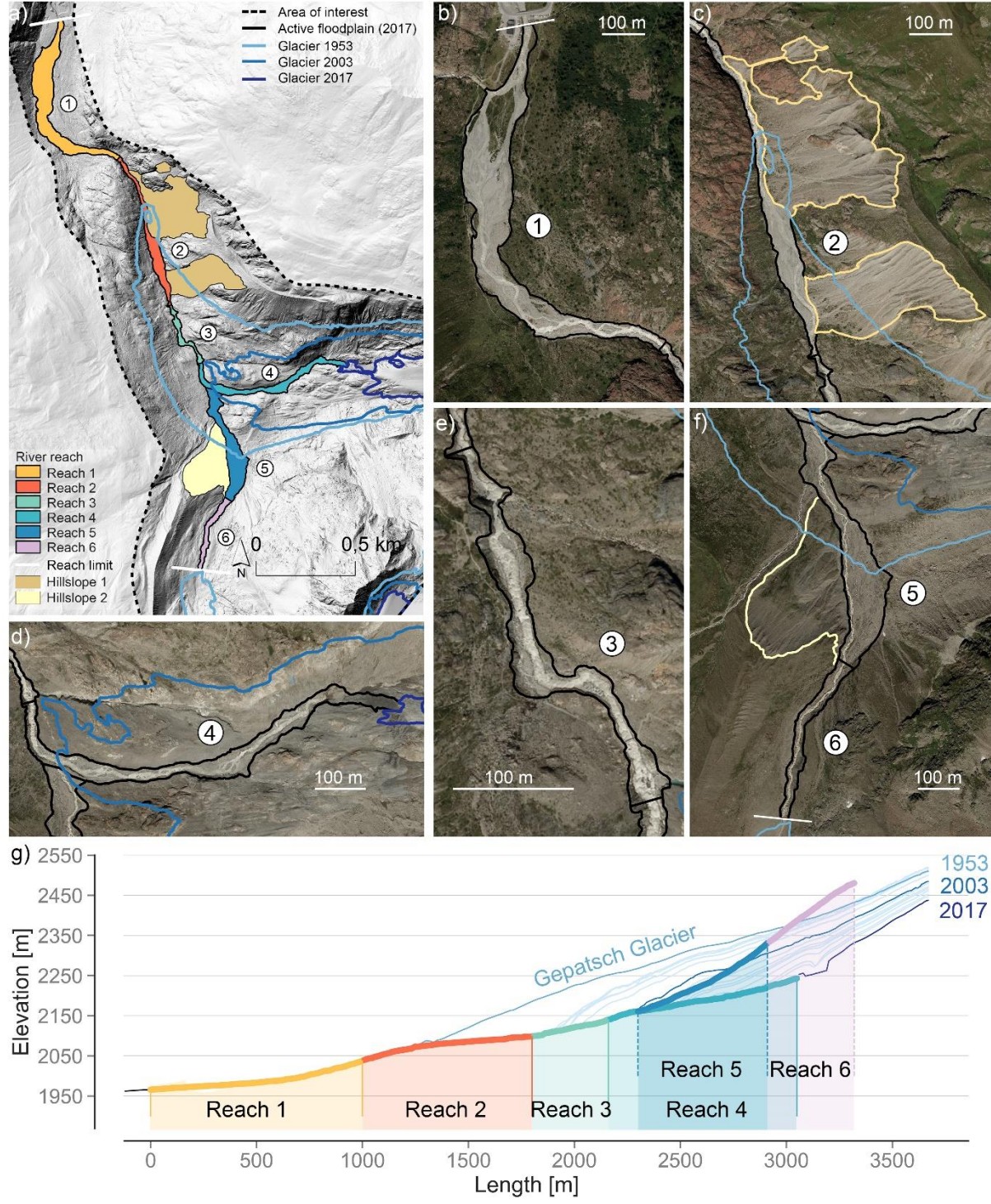

**Figure 2: Illustration of the area of interest with the mapped active floodplain and lateral hillslope as of 2017, the river reach delineation with their number and the glacier extent for the years 1953, 2003, and 2017 a). A zoom of the mapped morphologies**

superimposed to the generated orthophoto of 2015 is shown from b to f; the numbers correspond to the river reach number shown on the overview map. g) Longitudinal profile of the active floodplain centreline as of 2017 (see red line Fig. 1) coloured for each river reach and the elevation of the glacier for each time step.

The glacier outline mapping was done manually, with the visual support of the hillshade, the orthophoto when available, the LiDAR intensity raster, and the DEM of difference of consecutive DEMs. The latter was also used to detect areas of debris-covered ice on the glacier and exclude areas of dead-ice melt out on the glacier side and lateral hillslopes. The lateral hillslopes are mainly characterised by lateral moraines. The lateral moraine to the orographic right of the active floodplain (named hillslope 1) is coupled to reach 2 (Fig. 2c), while the lateral moraine (named hillslope 2) further upstream to the orographic left is coupled to reach 5 (Fig. 2e). The mean slope of hillslope 1 and 2 is 36° (max 84°) and 35° (max 72°), respectively. For mapping the contributing area of the lateral hillslope, the flow accumulation and watershed boundaries were used together with the orthophoto, DEM, and hillshade.

**Table 1. Characteristics of the river reaches as of 2017. Values refer to the active floodplain, and the slope is reported for the floodplain and active channel. Length is measured with respect to the centreline of the main channel (see Fig. 1). The six river reaches are shown in Fig. 2.**

| River reach no. | Mean elev. | Area | Max width | Min width | Max length | Mean slope floodplain (thalweg) | Mean slope floodplain (mapped) | Characteristics |
|---|---|---|---|---|---|---|---|---|
| | [m] | [m²] | [m] | [m] | [m] | [°] | [°] | |
| 1 | 1985.5 | 60991.0 | 112 | 25 | 1006.4 | 8.2 | 10.1 | Braided channel with mid/lateral un-vegetated and sparsely vegetated bars and vegetated island |
| 2 | 2080.9 | 23612.2 | 63 | 9 | 802.7 | 9.5 | 11.8 | Braided channel with mid-lateral un-vegetated bars |
| 3 | 2119.8 | 7903.6 | 35 | 10 | 358.0 | 12.2 | 26.4 | Confined and incised bedrock channel |
| 4 | 2189.7 | 26371.5 | 62 | 16 | 885.9 | 13.3 | 20.1 | Incised section with mainly bedrock bed |
| 5 | 2242.9 | 36496.6 | 111 | 27 | 592.1 | 20.1 | 21.9 | Braided channel with sediment bed |
| 6 | 2407.1 | 10778.5 | 42 | 18 | 398.8 | 25.5 | 31.5 | Incised channel with sediment bed |

## 2.4 Hydrological data

Discharge of the river Fagge in Upper Kaunertal has been measured since 1971. However, in the first years of installation, the data record contains many gaps and the time series became stable from 1977 onwards. The data are provided by Tiroler Wasserkraft AG (TIWAG). It is worth noting that this station is located downstream of the confluence of the Fagge and the Riffler Bach creek, and thus includes the discharge coming from both the Gepatsch Glacier and a second glacier in the catchment (Weissseeferner), which is not part of this investigation.

## 2.5 Analyses

We quantify the spatio-temporal changes of the proglacial active floodplain and we aim to identify the main drivers of channel sediment changes among the variables of glacier front variation, lateral hillslope erosion, and strong runoff events; these
variables are directly or indirectly influenced by climate change.

Given accurate co-registration of the DEMs, the DEM differencing method, i.e. the difference of two DEMs acquired at two different points in time, is used to quantify the spatial distribution of positive and negative differences (vertical elevation change) of the active floodplain. The DEM of difference (i.e. elevation change) is then used to quantify the volumetric change of sediment storage (Leyland et al., 2017; Vericat et al., 2017) in terms of deposition, erosion, and net volume for each time
step. Estimates of net change are based on unthresholded DEMs of difference (Anderson, 2019). Only the mapped active floodplain is considered for quantifying fluvial sediment volume change and sediment budget.

The active floodplain is divided into six river reaches (see Sect. 2.3) and within the river reaches, sub-reaches are delineated by clipping the active floodplain polygons with cross-sections drawn every 100 m along the thalweg. For each sub-reach, we quantify the net volume rate (i.e. the annual rates of volume change computed from the net volume divided by the duration of
the inter-survey period, m$^3$/yr) to identify potential spatial/temporal patterns in the net sediment budget of the respective reach. Subsequently, the volume rate of eroded and deposited sediments is also calculated for the entire active floodplain (i.e. all connected river reaches). Finally, from the temporal/sequential DEMs, the cumulative net volumetric sediment balance is calculated for the active floodplain as well as at the reach scale to define their contribution to the sediment budget.

While we do not know the amount of sediment from the glacier that enters the investigated river corridor (e.g. at reaches 4 and
6), we assume that the slopes of the lateral moraines directly adjacent to the Fagge at reach 2 and at reach 5 constitute major sources of sediment supplied to the river. In order to quantify this contribution, we computed the DEMs of difference for seven selected time periods. The use of fewer periods with longer durations (we aggregated shorter periods to resulting periods that were longer than one year) was necessary due to the lower data quality of the DEMs in the steeper hillslope sections compared to the river corridor.

The gauging station data are examined for discharge peaks in different periods. A trend analysis based on the non-parametric Mann-Kendall (Mann, 1945; Kendall, 1975) test is also carried out for the annual maximum discharge time series to identify the presence of statistically significant trend and to derive the linear regression line based on the Mann-Kendall test results (Hussain and Mahmud, 2019). The discharge values in the respective months are then plotted in order to classify them seasonally.

Analysis of volumetric changes based on high-resolution repeat topography (i.e. the DEMs) required an estimation of the uncertainty of the respective DEM (Anderson, 2019) which propagates into DEMs of difference and the volumetric changes computed from the latter. We quantify the elevation uncertainty using the DEM of difference between each DEM and the reference ALS DEM (2017) on stable terrain. The stable terrain is the same used for the ICP co-registration (see Sect. S1). The main statistics of accuracy and precision such as mean, standard deviation, and RMSE as well as robust statistics of the DEM

of difference such as median and sigma mad ($\sigma_{MAD}$)  (Höhle and Höhle, 2009) are calculated to assess the accuracy and precision of the generated DEMs. For normally distributed data, the $\sigma_{MAD}$ is defined as 1.4826*MAD, where MAD is the median absolute deviation. The $\sigma_{MAD}$ of the elevation change estimates on stable terrain is used as elevation uncertainty and is propagated into the assessment of the absolute sediment volume ($\sigma_v$). The following equation is used:

$$\sigma_v = nL^2 \sqrt{\sigma_{MAD\_DEMtime1}{}^2 + \sigma_{MAD\_DEMtime2}{}^2}$$

Where n is the number of cells used for the calculation, and L is the cell size in metres.

For the historical DEMs, we attempted to estimate the ranges at which errors are autocorrelated using the semivariogram and found a slight spatial autocorrelation with a range of a few tens of metres. However, since we are dealing with wider areas of both the active floodplain and the hillslopes that are large compared to the semivariogram range, the spatial autocorrelation was considered not to affect our error calculation. The errors in terms of mean and standard deviation between

temporal/sequential DEMs are also reported using the DEM differencing within their common stable terrain.

## 3 Results

### 3.1 DEMs of difference and uncertainty

For all the generated DEMs, the DEM of difference on stable terrain with the reference ALS after ICP co-registration shows impressive results in terms of accuracy and precision. Mean and standard deviations, as well as the $\sigma_{MAD}$ used for the error

propagation, are given in Table S3. The mean error around zero for both the LiDAR and digital aerial images DEMs shows that systematic bias is essentially removed with the co-registration. Slightly off the zero mean are the DEMs from the historical images. The largest error is recorded for the dataset from the 1980s and 1990s (mean error of about -0.08 m and -0.16 m for the 1983 and 1990 datasets, respectively). The $\sigma_{MAD}$ ranges between 0.22 (DEM 2003) and 0.41m (DEM 1982) for the historical images DEMs and it is below 0.16 m for the other DEMs. The different accuracies can also be seen from the 2D map

of the DEMs of difference in Fig. 3).

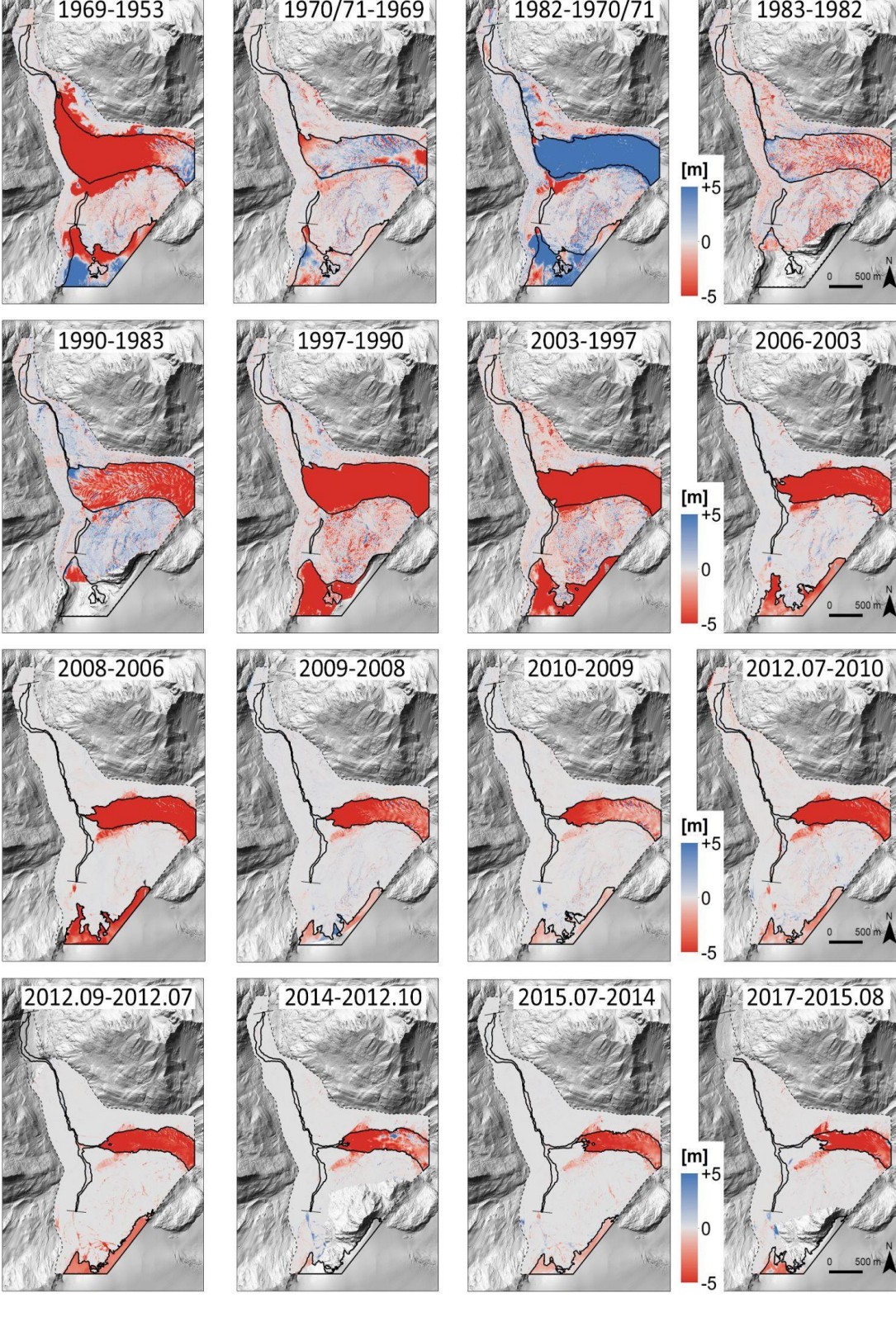

**Figure 3. Temporal/sequential DEMs of difference (i.e. the elevation change in metre) of the area of interest with the mapped glacier extent and the active floodplain. The elevation difference maps between 2019 and 2017, August 2015 and July 2015, October 2012 and September 2012 are not shown because at this scale, the monthly variation is very small and in 2019, only some parts of the active floodplain are reconstructed. In the background is the hillshade of the ALS 2017 DEM.**

## 3.2 Spatial change of the proglacial active floodplain in relation to glacier fluctuations

Glacier fluctuations are responsible for the variation of the area and length of the active floodplain (Fig. 4). The time series of DEMs of difference (Fig. 3) clearly highlight these fluctuations since 1953 as well as ice melting within the proglacial area. In the 1980s, the glacier experienced a period of advance preceded by a retreat of about 600 m between 1953 and 1969 (Fig. 4). A visible reduction of the glacier terminus occurs mainly after 2003 followed by an acceleration of the glacier shrinkage and retreat. With the glacier retreat, the northern active floodplain (reaches 1, 2, and 3) connected with the southern active floodplain (reaches 5 and 6) in 2003 and formed an additional channel branch (i.e. reach 4) to the east whose length and area continue to increase with the ongoing glacier retreat. As a consequence, in almost 70 years the area of the active floodplain increased by 60 % (Table S4), while the total length of the river measured at its centreline (Fig. 1, red line) increased from 1387 m (reaches 1 and 2) in 1953 to 4257 m in 2017, corresponding to a glacier retreat by about 1.8 km (Fig. 4). The widening of the active floodplain between the time intervals is also caused by lateral dynamics associated with erosion of the lateral riverbanks.

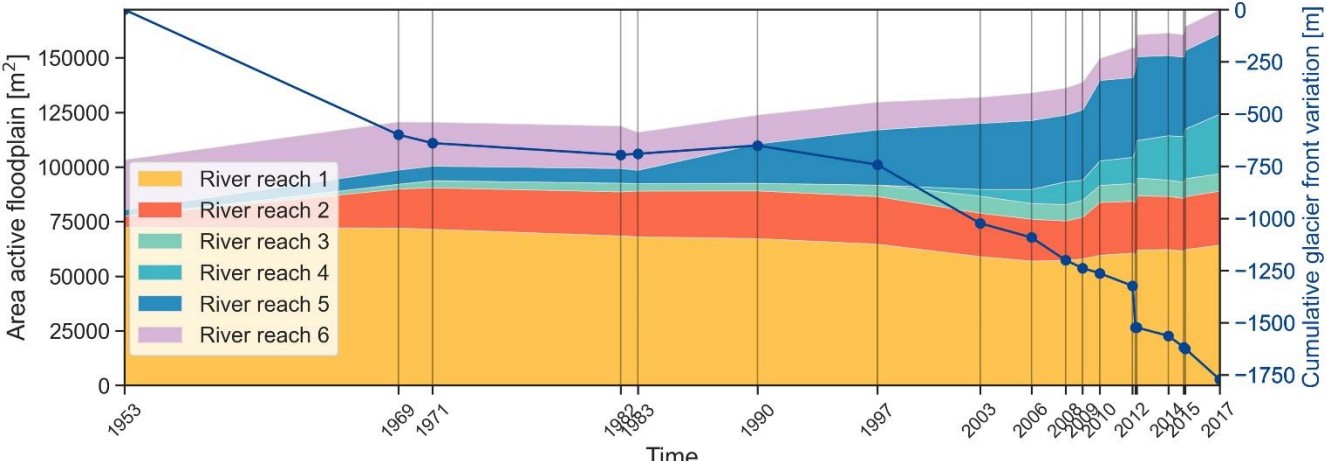

**Figure 4: Area change in square metres of the mapped active floodplain and the cumulative glacier front variation. This is calculated with respect to the river centreline as of 2017. Note that the year 2019 surveyed by UAV is missing as there is no data up to the glacier front.**

## 3.3 Spatio-temporal variation of the net sediment volume at the sub-reach scale

An overall, consistent spatio-temporal trend of the sediment storage at the sub-reach scale is not immediately detectable, besides the changes in the length of the active floodplain as a function of glacier terminus fluctuations (Fig. 5). However, some time steps do exhibit distinct horizontal (i.e. time) and vertical (sub-reach scale) patterns. From 1970 to 1982, a marked net erosion emerges in the middle of reach 1 (-19000 m$^3$), while further upstream (reach 2) an overall accumulation pattern can be

detected (~25000 m³). Similar sediment dynamics also appear between 1982 and 1983: a strong erosion (-3700 m³) in two sub reaches of reach 1, preceded by a marked accumulation on reach 2, which also occurred in almost the same sub reaches in the consecutive time step between 1983 and 1990. The entire active floodplain for the time step from 1997 to 2003 is characterised by erosion (-27100 m³, i.e. approximately -4500 m³ per year). Note that from 2003, reaches 5 and 6 are connected to reach 4. The subsequent time steps are shorter, and the two-year period between 2010 and 2012 emerges with a high variation of net accumulation downstream (reaches 4, 3, part of 2 and 1), counterbalanced by erosion and deposition in reach 5 and strong erosion upstream (reach 6). Between July and October 2012, the dramatic glacier retreat of approximately 200 m probably caused exceptional net erosion of 12000 m³ in the proximity of the glacier terminus (reach 4) with a deposition downstream (reach 2). The following time steps do not show strong volumetric change, except for higher accumulation in reach 1 (~3200 m³) between 2014 and 2015 and similarly between 2017 and 2019 (7800 m³).

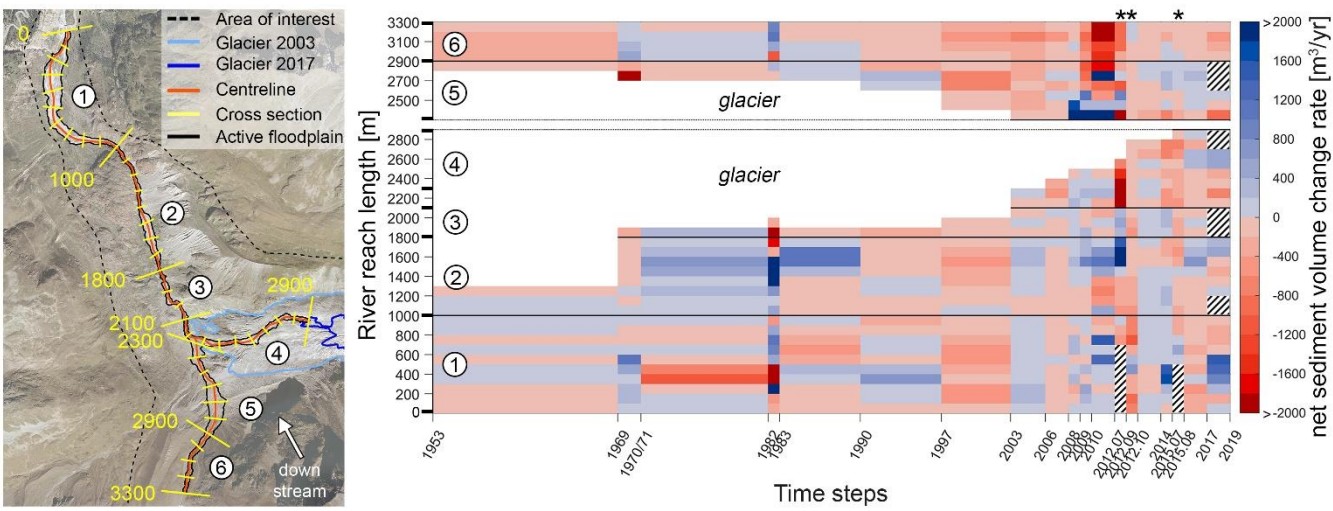

**Figure 5:** Spatio-temporal variation of the net sediment volume change rate of the active floodplain (reach 1 to 6) in cubic metres per year (m³/yr). On the left, a graphic illustration of the length of the reaches and the cross-sections drawn every 100 m along the thalweg superimposed on the generated orthoimage of 2003. The arrow indicates the downstream direction. Negative values (reddish colours) represent surface erosion while positive values (blueish colours) represent surface deposition. For better visualisation, a limit of 2000 m³ per year is chosen. Missing information is indicated with a hatched area. Note that for the monthly time step, indicated with * in the figure, the absolute volume change (i.e. m³) is reported instead of the volume rate. In addition, for better visualisation, their width corresponds to one year.

### 3.4 Sediment volume and net cumulative balance at the river reach scale

By computing the volume rate for each connected reach, a large amount of sediment mobilised between 2010 and 2012 in terms of both accumulation and erosion clearly shows up, which is also preceded by two years of high activity (Fig. 6a). For several time steps, erosion, and deposition are balanced. The highest accumulation value (40500 m³) is recorded between 1982 and 1983, while the extreme erosion event in the summer of 2012 reached about -32900 m³. This event is correlated to the collapse of the glacier front in the summer of 2012, and the corresponding peak of runoff up to 47.3 m³/s measured in August

2012. No extreme runoff events are measured before 1987, which represents the highest peak (59.2 $m^3$/s) of the recorded time series together with the runoff of 57.7 $m^3$/s in 2011. Overall, an increase in the frequency of extreme events (39.08 $m^3$/s, one standard deviation from the mean) can be observed. Indeed, further extreme events occur in 2014, 2018, and 2019 (Fig. 6c).

If we cumulatively sum up the net sediment budgets of the time steps for all connected reaches (black line in Fig. 6b), we can identify a trend in the net cumulative balance, and the contributions of different reaches (Fig. 6b). In connection with the glacier re-advance phase, which occurred between 1976 and 1988 as observed by Fischer et al. (2016), a consistent tendency of net aggradation is shown until the end of the 1980s. After a "plateau" with no increase in the total balance, a decreasing trend appears clearly since 1997 with a short increase between 2008 and 2010 to become negative in and after 2012. In the

summer of 2012, all river reaches show an abrupt negative jump (with a net volume change of more than -12000 $m^3$, Table S4) except an accumulation in reach 2. The overall negative balance seems to be governed mainly by reaches 6, 5, and 4 (after 2012). Reach number 5 is responsible for the short positive bump in the net cumulative balance between 2008 and 2012, while reach 6 shows a steady degradation tendency after 1997. On the contrary, the downstream reaches 1 and 2 have an almost persistent aggradation trend after 2003. The lower sediment contribution of reach 3 appears again due to its smaller area and

confined morphology that does not provide accommodation space for sediment. Analysing the cumulative net sediment balance normalised by the area of the floodplain (i.e. the elevation change) (Fig. 6c), a similar trend emerges, but the cumulative net balance of all connected reaches shows less variation. In addition, the negative contribution of reach 1 is less dominant due to its large area. Reach 6 shows the highest amount of erosion with a cumulative elevation change of up to -5 m followed by reach 4. Interestingly, all reaches show a stable trend toward aggradation after 2012. The volume of erosion, deposition, and

the net for each river reach is illustrated in Figure S4, and values are reported in Table S4.

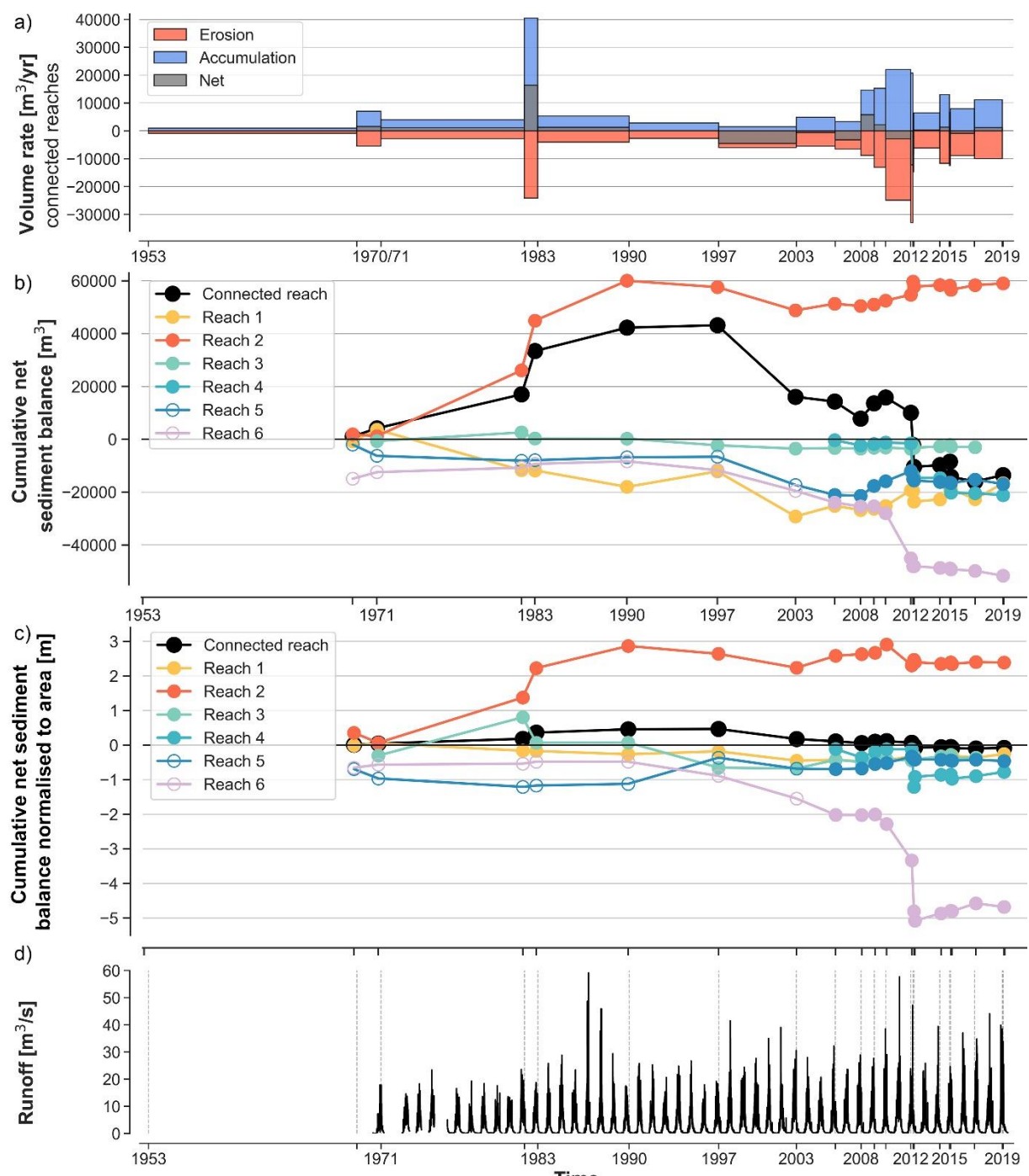

**Figure 6: a) The volume change rate and b) the cumulative net sediment balance in cubic metres and c) in metres (i.e. normalised to area) for the connected reaches and for each river reach. Note that following the topology of the reaches and the time of their deglaciation, the net balance for the first time step contains reaches 1 and 2 only, reach 3 is included from the second time step onward, and reaches 4, 5, and 6 are contained in the balance from 2003 onward. The cumulative balance for the reach 5 and 6 is**

also reported for the first time stamps (until 2003) as open circle because they did not contribute to the total budget (full circle) until 2003. d) The recorded runoff data with marked time steps of the available DEMs as a dashed line.

### 3.5 Sediment volume estimation of connected hillslopes and runoff analysis

Looking at the net volume rate of the lateral hillslopes, both are characterised by a large erosion activity in the period between
1997 and 2003 (Fig. 7a). However, we assume that over the same time interval, the eroded sediments were not deposited in the connected river reaches as these also show a negative volume rate. Hillslope 1 shows a large amount of sediment transferred to the connected river reach 2 in the 1980s (Fig. 7b), but after 2003 its contribution declines almost exponentially (Fig, 7a). The large aggradation in the river reach 5 (9400 $m^3$) between 2008 and 2012 (July) coincides with a net erosion of the coupled lateral hillslope of about -15000 $m^3$ (Fig. 7, a and c) in addition to a strong erosion in the upstream river reach 6 (~-19800 $m^3$) over the same period. Sediment volume estimation for selected time steps of hillslope 1 and hillslope 2 are reported in Table S5. The glacier is the other main source of sediment to the active floodplain, although it cannot be measured directly. However, runoff as a measure of meltwater from snow and glaciers can be used as a proxy for the sediment source. The maximum annual runoff measured at the gauging station shows a statistically significant (p-value of <0.01) increasing trend of 0.35 $m^3$/s year$^{-1}$ and a total increase of 15 $m^3$/s over the entire period from 1971 to 2019 as observed by Altmann et al. (2020) (Fig. 9b). The four highest extreme events occurred in August, but in general, the highest mean discharges occurred in July (Fig. 8a), which often corresponds to the time of the topographic data acquisition. The re-advance phase of the glacier (until 1986) coincides with a rather low recorded runoff (19.0 $m^3$/s) despite the strongest event recorded in 1987. The time step between 1990 and 1997 is also characterised by a low mean maximum annual runoff of 22.2 $m^3$/s compared to 31.1 $m^3$/s between 1997 and 2003 (Fig. 8b).

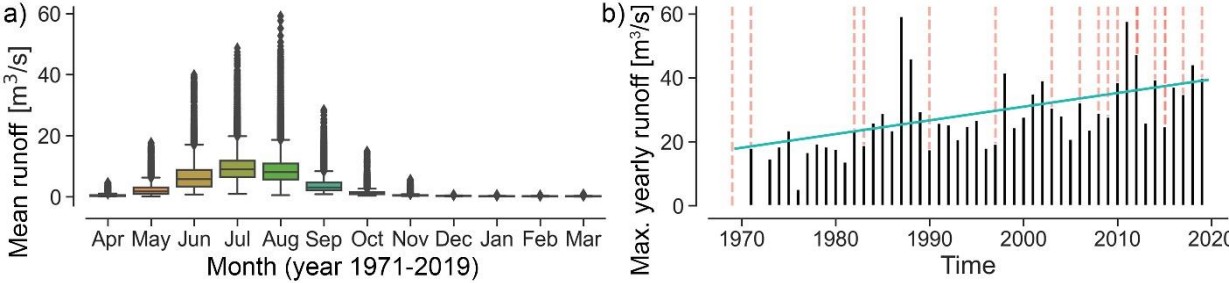


**Figure 7: a) The rate of net volume change of lateral hillslope 1 and 2 (reference to Fig. 2) for the selected time steps and the corresponding net volume change rate of the active floodplain of the river reach 2 and 5, respectively. Note that sediment from hillslope 2 is transferred to reach 5 from 1982 onwards. For the selected time steps, the elevation difference map b) of the lateral hillslope 1 and c) the later hillslope 2, with the mapped active floodplain and glacier.**


**Figure 8: The runoff (m³/s) between 1971 and 2019. a) The mean yearly runoff events of the respective months and b) the maximum yearly runoff (m³/s) of each year and the linear trend. The vertical dashed lines mark the time of available DEMs.**

## 4 Discussion

### 4.1 Data quantity and quality

There is a large archive of stereo aerial photographs from film and digital cameras acquired in Austria (available in https://lba.tirol.gv.at/public/karte.xhtml) between the 1940s and 2010s, containing valuable information on geosystem changes. This opens the possibility of creating a unique time series of elevation changes in Alpine catchments.

Our study analyses past and recent proglacial river sediment changes of the main channel network of the Kaunertal Alpine catchment located in Austria using all available historical and recent aerial imagery for orthophoto and DEM generation,

airborne LiDAR DEMs (many of which were acquired with the aim of geomorphological analysis) and datasets generated from drone flights. Our dataset covers 66 years, spanning nineteen periods from 1953 to 2019 representing a unique dataset in terms of its length, temporal resolution (inter-survey periods between one month and 16 years), and high spatial resolution and accuracy. DEMs from historical images were generated with a resolution of 1 m with an accuracy of less than 0.4 m ($\sigma_{MAD}$); DEMs from digital images and LiDAR with a resolution of 0.5 m show an accuracy of approximately 0.2 m.

The generation of such high spatial and temporal resolution datasets in other glacial catchments in the European Alps or in other high-mountain areas worldwide is possible, but the data availability is a significant challenge. Concerning historical aerial images, there is a lack of overview of aerial image archives with regard to their spatial and temporal coverage. In addition, many archives of historical aerial images are not freely accessible and images collected for military purposes have yet to be declassified or have restricted access. One additional problem is the image quality, which is often compromised by

photo processing, physical storage conditions, and the digitisation process, i.e. the distortion introduced during the scanning and the resolution of the scan (Stark et al., 2022). Furthermore, external information such as camera calibration certificates required for accurate photogrammetric processing, is often missing. Nevertheless, automated methods using SfM are currently being developed to process historical aerial images and generate time series DEMs and high-resolution orthoimages (Knuth et al., 2023). Images from the declassified stereo reconnaissance satellites from the Corona (Dehecq et al., 2020; Ghuffar et al.,

2023) and Hexagon KH-9 missions can be easily accessed via the USGS Earth Explorer portal (free of charge for images that have already been scanned) and offer great potential for DEM reconstruction for geomorphic change detection (Maurer et al., 2015). However, despite their global coverage, their period of acquisition ranges between the 1960s and 1980s and their ground resolution from approximately 2 m to 8 m. From the year 2000 on, private companies have often conducted airborne digital photogrammetric and Lidar surveys, but rarely make the data available. Modern high-resolution stereo satellite images with

metre to sub-metre resolution (e.g. Worldview and Pléiades) are commercially available after 2005 and can be used for large-scale DEM reconstruction, but they are not freely available and have limited temporal resolution. Consequently, studies in Alpine catchments with high spatial and temporal resolution datasets dating back to the last century are still very limited, preventing the comparison of spatio-temporal variation in sediment dynamics and the assessment of the response of catchment-scale sediment yield to climate change.

From temporal/sequential DEMs, changes in elevation and thus the volume change in terms of erosion, deposition, and net
       sediment balance of the active floodplain and the coupled lateral hillslope can be calculated. The net balance of the hillslope
       sections directly translates into the sediment input to the downslope river reaches. However, negative volumetric change due
       to the melting out of dead ice must be excluded from the calculation, for example using the spatial pattern of the DEM of
       difference for the detection (e.g. Fig. 7b, period 1971-1953) (Betz et al., 2019). Furthermore, we have not considered the water
depth when assessing the sediment volume on the active floodplain. In fact, with the technology used, both photogrammetric
       and LiDAR DEMs provided data for the exposed areas of the riverbed and the water surface morphology. The water surface
       height of Alpine rivers in summer (i.e. glacier melting season) depends largely on the time of day and the general weather
       before and during the day of acquisition. However, this information is not available and we cannot assume the data were always
       acquired at low flows. Therefore, we compared the net volume for each river reach and for the whole active floodplain from
selected LiDAR DEMs with or excluding the active channel (i.e. the water surface) (Fig. 9). The comparison shows different
       results at the reach scale and for the whole active floodplain. However, this disagreement can be related to the large water area
       (up to 60 % e.g. river reach 2) within the active floodplain. In fact, where only 30 % of the active floodplain is covered by
       water (e.g. river reach 6), volume calculations are comparable (Fig. 9). Therefore, excluding the area covered by water from
       the sediment balance calculation would also introduce an error, as we would exclude the change there. Baewert and Morche
(2014) also mentioned this issue, concluding that in the case of a braided system, the shift of the river course between different
       surveys can compensate for this error to some extent, but this depends on the area covered by water. Bathymetric reconstruction
       of the riverbed from LiDAR and photogrammetric DEM is still a subject of research, and most studies of river sediment based
       on DEM of difference do not exclude the water area in the calculation (e.g. Calle et al., 2020; Scorpio et al., 2022; Savi et al.,
       2023). Only a few studies include the refraction by water during SfM photogrammetric reconstruction of the river surface
(Dietrich, 2017, Lane et al 2020). This development should be considered when UAV-based SfM photogrammetry is applied
       to derive erosion and deposition analysis of river environments from DEM, but it requires clear water conditions (Bertalan et
       al., 2023) that are rare in a glacier-fed stream. Moreover, it is unclear whether the bathymetric correction works as well for
       historical aerial photographs as for modern high-resolution UAV-based imagery.

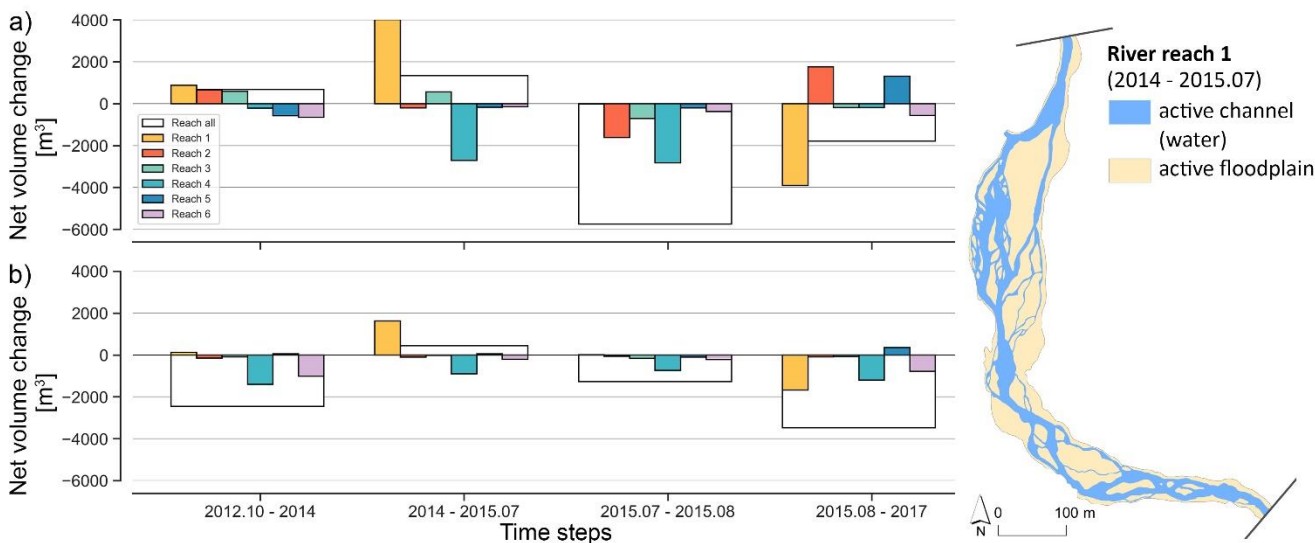

Figure 9: Net sediment volume change of the active floodplain for each reach and all connected reaches a) with and b) without considering the water surface in the calculation. On the right, the active floodplain and the active channel (i.e. the water surface, which area covers about 41 % of the active floodplain) is illustrated for reach 1. The mapping of the active floodplain and active channel for the other river reaches is shown in Figure S5 for the time step 2014-2015 July.

## 4.2 Sediment balance at reach scale and sediment source

Looking at our sediment balance data, we have to stress two main caveats. First, the data mainly refer to coarse sediment as the fines are transported in suspension and can be removed or deposited without major surface changes that are detectable on (especially the historical) DEMs of difference. Second, we can only report net changes to the sediment storage as information on the sediment fluxes is not available; for closing a proper coarse sediment budget, we would need to know either the amount of sediment leaving the study area or of sediment supplied by the glacier in each time steps. Research on the contribution of glaciers to the sediment budget of proglacial areas is scarce and reported results display considerable variance. For example, Leggat et al. (2015) state that circa 60% of suspended sediment load in a proglacial river is derived from the glacier, while O'Farrell et al. (2009) found that non-glacial sources contribute as much as 80% (± 45%) to proglacial sediment yield in their study area.

The general picture that emerges from the cumulative sediment balance is an increasing trend until the 1980s, followed by a steady balance until the late 1990s, then a systematic decrease interrupted by a short increase between 2008 and 2010. It followed a rapid decrease and a shift towards a negative balance to end with a stable negative balance (Fig. 6).

The proglacial river in our study is in transition with massive glacier retreat creating new channel networks (Fig. 2b), with changes in runoff (increasing according to the gauge record) and probably in sediment transport capacity, and possibly a decline in sediment supply, which leads to a progressively negative cumulative sediment budget for the whole study area. The systematic decrease in the cumulative sediment budget began between 1997 and 2003, which can be attributed to the drastic glacier shrinkage and retreat, as also observed by the available measurements of the front variation (WGMS 2021) and the

high runoff recorded. Furthermore, a large erosion activity also characterised the lateral hillslopes (Fig. 7) that are/were coupled to the channel network and therefore contribute to sediment supply. Extreme runoff events also lead to large erosion upstream. However, in order to better understand the links between trends in sediment storage and different sources and processes, information on the location of possible sediment sources (by aggregation), but also on their condition/availability of sediment (bedrock vs. fluvial deposition) and their accessibility (no "accommodating" space for sediment) needs to be analysed at the river reach scale.

We can clearly see the influence of the "accommodation space" offered by braided systems on the storage changes. River reach 2 is characterised by aggradation over almost the entire period due to the availability of ample accommodation space (and widening of the braid plain). There we see massive deposition when the sediment load is high, and the overall (cumulative) balance is positive to stable. This occurs during an almost stable phase of the terminus position of the Gepatsch glacier between 1969 and 1997, including an advance phase that started in 1976 until 1988 as already observed by Fischer et al. (2016). Furthermore, the connected deglaciated lateral hillslope exhibited the highest erosion activity on its ridge, with approximately $\sim$-55000 m$^3$ of sediment being transferred to the river reach between 1983 and 1982. This sediment transfer probably contributed to local aggradation in river reach 2 ($\sim$19000 m$^3$, Fig. S4) and further downstream in reach 1. However, the braided river reach 1 shows a localised erosion pattern (Fig. 5, reach 1) and a negative net balance until the 1980s due to human disturbances such as road and bridge construction as evidenced by historical orthophotos (Fig. 10a).

The presence of bank collapses in both braided river reaches suggests that they have a high degree of lateral mobility, which, in the case of river reach 2, promotes a direct coupling (i.e., structural connectivity) between the processes occurring on the lateral hillslopes and those taking place within the active floodplain, as noted by Savi et al. (2023). The widening of the active floodplain caused by lateral dynamics has been identified as an important factor of lateral connectivity and sediment supply (Cienciala et al., 2020).

The river reach 3 is quite narrow, confined between rocky slopes, and therefore does not have much accommodation space. This explains the near-zero changes in sediment storage: there is only a little storage as the sediments eroded upstream are flushed through, representing a source for the aggradation of river reach 2.

The river reaches 4, 5, and especially 6 give a negative impulse to the cumulative sediment curve indicating that the sediment storage is depleting there. We do not know the amount of glacial sediment that enters the investigated river corridor, e.g. at reaches 4 and 6, however, these two reaches show different sediment trends. We can clearly see that river reach 6 functioned as an ample sediment source due to the large availability of fluvial sediment. On the contrary, reach 4 in recent years does not experience much erosion as it is mainly characterised by bedrock, as can be deduced from the generated orthophotos and DEMs (Fig. 10c). However, in the summer of 2012, erosion occurred in the channel network itself with a major channel incision and a subsequent formation of a scarp at the left boundary of the river corridor where the river reach 5 joins reach 4 (Fig. 10c) as reported by Baewert and Morche, (2014). On 26 August 2012, a heavy rainfall event occurred (Baewert and Morche, 2014, Hilger et al 2019), which presumably triggered a subglacial water pocket outburst, already anticipated by the circular ice pattern visible in the July 2012 LiDAR DEM (Fig. 10b). The high-magnitude event (maximum discharge 47.3 m$^3$

s$^{-1}$) caused channel incision, and a total erosion of more than 30000 m$^3$, which resulted in the transition to cumulative negative sediment storage. During this event, all river reaches experienced a decrease in sediment balance, while systematic aggradation continued to characterise the braided river reach 2 (~5000 m$^3$). In contrast, in the work of Baewert and Morche (2014) in the Fagge River, aggradation dominated all but one of their five river reaches proximal to the glacier with a volume change of about -69000 m$^3$ between June 2012 and September 2012 with erosion of up to 5 m in the bed channel. The reason for this discrepancy is certainly related to the different datasets (and accordingly different inter-survey periods and different spatial units) used. . Baewert and Morche (2014) worked on disconnected smaller river reaches, while we derived the balance from all connected river reaches. The discrepancy also points to the high dynamics nature of proglacial areas where even large short-term changes can be compensated on the medium to long term, and underlines the dependence of results on the spatial and temporal scale (in terms of extent and resolution) of such an investigation.

River reach 5 governed the positive bump in the sediment balance, which occurred between 2008 and 2012 (July). The aggradation trend of reach 5 in this period is explained by the sediment supply from the upstream reach 6 as well as from the connected lateral hillslope (erosion rate of about 4000 m$^3$/yr), probably triggered by heavy rain events - in the summer of 2011 occurred the second highest runoff event ever recorded.

After 2012, the overall cumulative sediment balance is rather stable despite an increase in the runoff trend and strong events. This can possibly be explained by a decline in sediment supply due to a transition from a sediment bed to a bedrock (Fig. 10c, 2019). Similarly, Delaney et al. (2018), found reduced sediment availability in a proglacial area of a Swiss catchment in recent years, probably attributed to further stabilisation of gullies in the proglacial area, along with the removal of transportable sediment from the area. Carrivick et al. (2018) discovered that lithology, thus grain size, plays a key role in determining the topography of proglacial systems, the spatial pattern and the link between hillslope and fluvial processes, ultimately affecting sediment supply, and transport.

Despite the observed similarities, comparing our sediment balance results in terms of cubic metres to other glaciated Alpine catchments is very challenging due to the complexity of the mountain environment and the high dynamics nature of the river systems in proglacial areas, which are subject to different forcing and with varying lengths of deglaciation periods. As Carrivick and Tweed (2021) noted, specific sediment yields from glacierised catchments can vary widely on a global scale, ranging from dozens to thousands of tons per year and square kilometre in the European Alps. In addition, differences in the dataset and survey period may introduce another limitation to comparability between the study areas (see above). However, if data of comparable quality and spatial/temporal resolution are available, it is possible to identify site-specific developments and/or common trends, as well as general processes affecting the proglacial area and Alpine catchments.

The emergence of new glacial forefields previously ice-covered is the first evidence of unprecedentedly rapid glacier melt caused by rising temperatures and longer heat waves that are affecting all Alpine glaciated catchments worldwide. Carrivick et al. (2013) and Baewert and Morche (2014) in their studies show that erosion is the dominating process taking place in the proximal area of the glacier, and accumulation generally occurs in the distal area. Similarly, Anderson and Shean (2021) in their study of proglacial erosion rate found that exported materials tended to accumulate in large deposits below the proglacial

limits, to then be distributed over subsequent decades or centuries. However, we show that fluvial sediment storage varies considerably depending on factors such as the local topography of the newly exposed active floodplain (bedrock versus sediment glacier bed, confinement versus presence of "accommodation space"), increasing floodplain area with the formation of new channels, sediment connectivity, and the subsequent sediment source from the lateral moraine. Retreating glaciers may uncover clean bedrock or may expose large sediment sources stored in their lateral and ground moraines. The amount of these

glacigenic sediments depends on the sediment balance of the retreating glacier, which is a function of the sediment production of the rock wall erosion and the erosional potential of the glacier runoff, i.e. slope, balance, and precipitation (Zemp et al. 2005). In addition, a single localised event such as an extreme precipitation event or the outburst of a subglacial water pocket can cause massive changes in the channel system, as also concluded by Anderson and Shean (2021), and numerous studies on the European Alps (e.g. Baewert and Morche, 2014, Lane et al. 2016, and Carrivick et al. 2013; 2018) and Himalaya (Cook et

al., 2018). Furthermore, Buter et al. (2022) and Savi et al. (2023) note that intense rainfall events play a critical role in promoting and enhancing functional sediment connectivity (Wainwright et al., 2011) among landforms that are already structurally connected. In addition, extreme events can modify the structural connectivity by removing barriers to sediment fluxes (Turley et al., 2021).

In their recent review, Zhang et al. (2022) present a global inventory of increases in erosion and sediment yield resulting from

535 cryosphere degradation. Their findings suggest that sediment transport will continue to increase until it reaches a maximum (sediment peak). This trend is linked to alterations in seasonal water supply, including increased winter liquid precipitation, early snowmelt and ice melt, and reduced snow-melt supply in later summer months, as also stated by Lane and Nienow (2019). Huss & Hock (2018) also noted that the "peak water", i.e. the maximum runoff from glacier long-term storage, will be reached in the coming decades due to ongoing glacier shrinking. All these factors will affect the hydrological and

540 geomorphological processes of glaciated catchments with consequences for ecological functioning of Alpine streams, water-related hazards, downstream water availability, sediment management in large reservoirs, and hydropower production (Schaefli et al., 2019). With regard to sediment management strategies, each site should be analysed separately as the mobilisation of eroded sediment downstream into the reservoir, and sediment transport in Alpine river systems in general, is complex and varies between catchment and landform types (Savi et al., 2023). Our analysis revealed a negative sediment

balance that requires consideration in the management of sediment deposits. However, we identified that braided systems (e.g. river reach 2) offer ample accommodation space, where sediment derived from the (limited) upstream fluvial accumulation could be deposited. Additionally, after the "peak water", there is likely to be an increasing lack of transport capacity to carry sediment downstream into the reservoir. A comparison of the bathymetric data of the reservoir may provide a more accurate estimate of total deposition of eroded sediment in the reservoir with respect to the volume changes from the proglacial area.

Overall, predicting sediment dynamics in a warming world is not yet well established and requires further research on integrating sediment observations from multiple sources, developing sediment-transport models, and enhancing interdisciplinary and international scientific collaboration (Zhang et al., 2022).

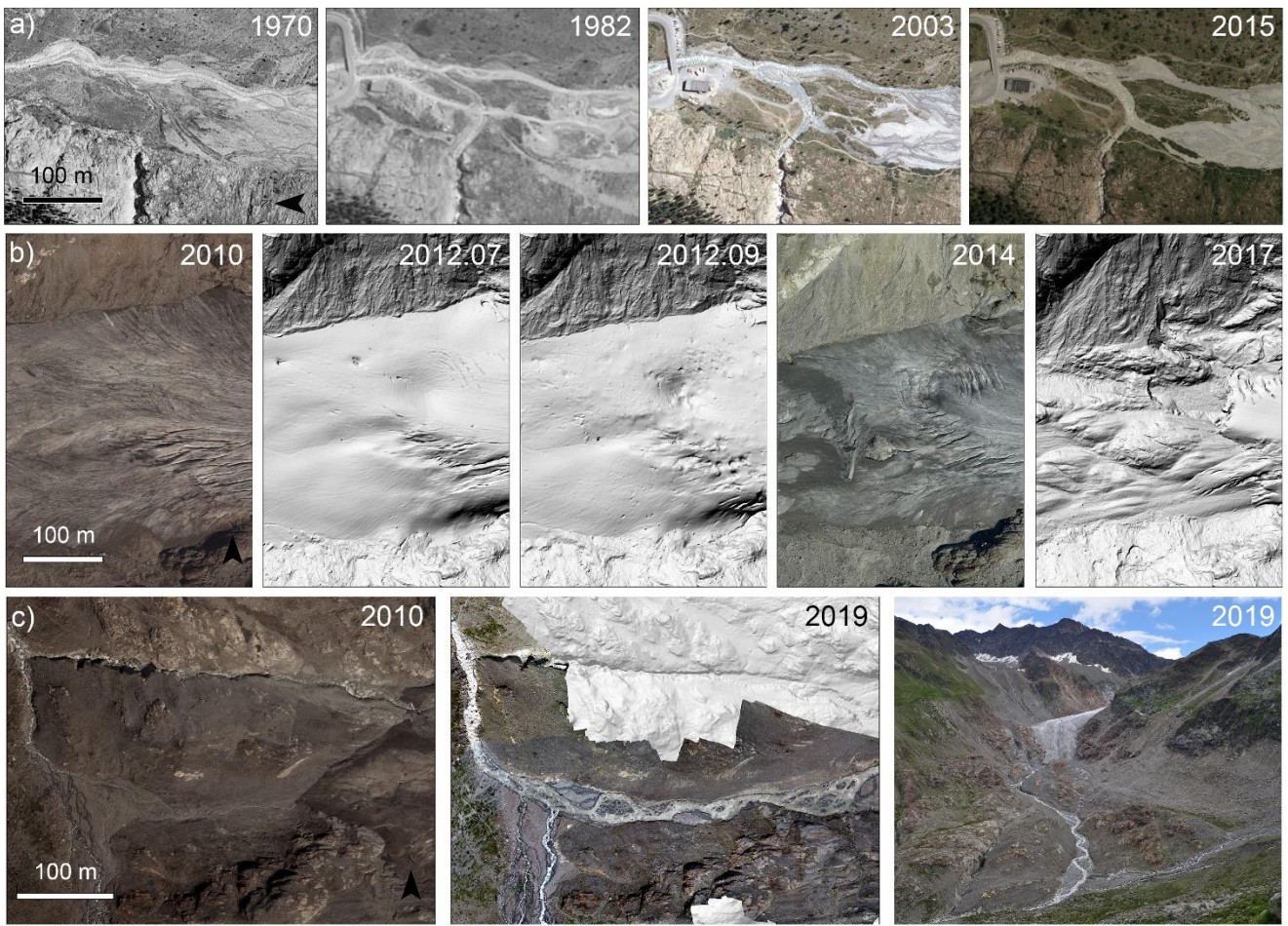

**Figure 10: Orthophotos and hillshade DEMs of selected time steps that visually show a) the human activities on the river reach 1 downstream after 1970; b) the development of a circular pattern (2012 July) on the glacier and the outburst of subglacial water pocket and c) the proglacial area change over 9 years such as channel incision and rocky characteristics of the river bed.**

### 4.3 Impact of the survey period on the sediment balance

Our study shows that whilst we are not able to capture every geomorphological event in the proglacial area, we are able to measure the aggregate effects of geomorphological activity between surveys and explain the main factors influencing the sediment balance. In 66 years, we have approximately -15000 cubic metres of net sediment storage. However, the longer the inter-survey period, the greater the likelihood of cancelling out the effect of a large event as can be seen in Fig. 11. The lowest rates of changes occur in the longest survey period (Fig. 11), which may be an indicator of the influence of the length of the period on the net results, as successive stages of erosion and deposition are more likely to compensate for each other than in shorter intervals. Furthermore, missing high-magnitude low-frequency events can lead to significant underestimations of long-term sediment delivery (Carrivick et al., 2013). In our case, by increasing the time steps, the trend does not change significantly. However, the total cumulative net balance is underestimated by about 5000 m$^3$ by deriving the volume before or after a strong

event (2012 July vs 2012 October), and there the trend changes. In fact, fluvial sediment transport is characterised by a very high temporal variability, and the survey frequency on calculated erosion and deposition volume has an impact (Milan et al. 2009). Therefore, high temporal resolution data are important to detect the variability of the proglacial river sediment budget. In recent times, this can be achieved using drone surveying. However, in addition to the limited spatial coverage of drone surveys, their operational use over the glacier forefield will become more difficult as glaciers continue to retreat and the proglacial area expands.

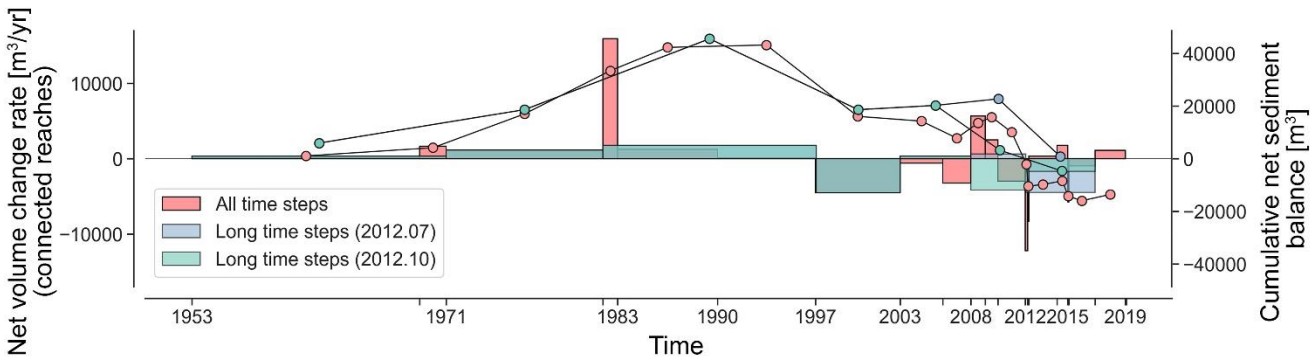

**Figure 11: Net volume change rate (bar plot) and cumulative net sediment balance (dots) for the active floodplain for all time steps including monthly data (19 DEMs) and for longer time intervals (7 time steps, with a minimum interval of 4 years). For these time steps, we show the difference in the cumulative net sediment balance when the interval is between 2008-2012 July (before the heavy rainfall event) and between 2008-2012 October (after the heavy rainfall event).**

## 5 Conclusion

This study presents 66 years of sediment changes of the proglacial Fagge River in the Kaunertal catchment, Austria, using multiple sources of historical and digital images and LiDAR data. Up to 19 periods from 1953 to 2019 spanning inter-survey periods between one month and 16 years are analysed through high-resolution DEMs - this long period is the most valuable part of this study. This allows us to identify periods of different sediment budgets and furthermore to attribute this behaviour to glacier front variation, lateral hillslope activity and runoff events as well as to the location of possible sediment sources and the availability of sediment (bedrock vs. fluvial deposition).

The general picture of the sediment balance is a consistent tendency to aggradation and balance until the late 1990s probably driven by the glacial advance (ended in 1988), and the sediments supplied by the lateral hillslope transferred to the connected braided system. The massive glacier retreat in the 2000s linked to the increasing runoff coincides with the starting of degradation tendency of the river sediment balance, which is mainly governed by the upstream river reaches. In fact, the braided system downstream shows an almost persistent aggradation trend throughout the entire investigation period as it provided accommodation space for sediment. Erosion activity on the lateral hillslope decreased drastically after 2003 with a few exceptions of sediment transfer during the occurrence of heavy rainfall events. The two strong runoff events after 2010

led to an acceleration of the degradation trend, which turned negative in 2012 after a sub-glacial outburst. In the end, we have approximately -15000 m$^3$ of net sediment storage in 66 years.

The continuous retreat of the glacier to higher elevations showed exposed bedrock substrate suggesting that the last stable trend of the river sediment balance may be explained by the reduced availability of sediment.

Overall, our study demonstrated that DEMs from historical images exhibit the capability in capturing continuous erosional and depositional patterns at 1 m resolution. These data represent a unique source to reconstruct past elevation changes and thus sediment-related processes in Alpine catchments. Moreover, we can show that short, high-magnitude meteorological and
hydrological events associated with local glacier retreat have a huge impact on the sediment budget. Therefore, high temporal and spatial resolution data are required to detect the variability of the sediment budget of a proglacial river. This can currently be obtained using UAV or very high-resolution satellite stereo imagery.

**Author Contributions**

Planning and conceptualisation were done by LP and TH. LP wrote and revised the article and prepared all graphics and tables.
LP created all the maps and orthophotos from the data used in this study. TH reviewed the content, offered substantial improvement, and played a strong role in the scientific supervision and the refinement of the focus of the paper. LP processed the photogrammetric data from historical images, and aerial and UAV images, and post-processed and analysed the topographic data. CR processed selected data from historical images and gave great support with the photogrammetric knowledge. LP, MS, FF, AM, and JR contributed to field data collection of TLS and UAV; MA processed the TLS data. MS
contributed to the UAV post-processing. SBN contributed to the discussion and related analysis. FH processed the ALS data and MW did the ALS strip adjustment. All authors have reviewed, read, and agreed to the submitted version of the manuscript. MB, FH and TH were responsible for funding acquisition.

**Funding**

This research is supported by the German Research Foundation (DFG) (project number 394200609; grant numbers BE
1118/38-1, BE 1118/39-1, BE 1118/40-1, HA 5740/10-1 and HE 5747/6-1) and by the Austrian Science Fund (FWF) (grant number 4062-N29).

**Acknowledgments**

This study is part of the SEHAG (SEnsitivity of High Alpine Geosystems to climate change since 1850) research project, funded by the German Research Foundation (DFG), the Austrian Science Fund (FWF), and the autonomous province of South
Tyrol and by the Swiss National Science Foundation (SNF). We thank the Federal Office of Metrology and Surveying (BEV)

and the Province of Tyrol (Land Tirol) for providing all the essential data. We thank Michael Zemp for inspiring discussions on glacial and periglacial sediment balance. The authors also thank the associate editor and two anonymous reviewers for their thorough and constructive feedback on our manuscript.

## Data Availability

The data used in the present study are available upon request from the University of Eichstätt-Ingolstadt within the framework of the SEHAG project (https://sehag.ku.de).

## Conflicts of Interest

The authors declare no conflict of interest.

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
