# Peer review of "Evolution of an Alpine proglacial river during seven decades of deglaciation"

_Earth Surface Dynamics, 2022_

## Author Comment (AC1)

*Dear Reviewers,*

*Thank you for your comments. We revised the manuscript considering all the reviewers' feedback. We improved the abstract by highlighting the main findings of the study. As suggested, we clarified in the introduction the main hypotheses that guided the work. We have also substantially expanded the discussion section, including a discussion on the feasibility of generating similar datasets, a comparison with previous studies, as well as addressing all the points raised in the introduction such as sediment connectivity and 'peak water'. We also improved the figures based on reviewer feedback and updated the reference list.*
*Below are all the point-by-point responses to the reviewer's comments.*

**Response to the reviewer's comments**

**RC1: 'Comment on esurf-2022-63, Anonymous Referee #1**
The presented work provides a significant contribution to quantitative research on fluvial sediment changes by estimating the sediment balance of a main Alpine river (Fagge River, European Alps) in a glaciated catchment system (Kaunertal in Austria), using multiple sources of historical and digital images and LiDAR data. Nineteen survey periods from 1953 to 2019 spanning inter-survey periods between one month and 16 years are analysed by using high-resolution DEMs. The analyses allow for identifying periods of different sediment budgets and for relating detected changes to glacier front variation, lateral hillslope activity and runoff events as well as to the location and activation of possible sediment sources.
This work can certainly contribute to solving one of the key problem of existing current studies, which is the lack of information over a longer time period. Process monitoring efforts are usually restricted to a few decades (at best). The material is very well presented and the manuscript is in all parts very well written. The manuscript is in my eyes excellent and has no significant flaws.

*--- We thank you for your positive feedback and for the comments!*

However, I have one issue the authors might consider: The authors highlight that their detailed analyses are built on a unique dataset. Referring to this point, I would like to ask if some more critical discussion on the potential of using the selected approach also in other study areas could be added. How likely is it to carry out this type of in-depth study in a successful way also in other glaciated catchment in the European Alps and in other high-mountain areas worldwide?

*--- Thank you for the suggestion and we added in the discussion section a paragraph on the feasibility of generating similar datasets.*
*"It is feasible to generate high spatial and temporal resolution DEMs in other glaciated catchments in the European Alps or in other high-mountain areas worldwide, but the lack of available data is a significant challenge. Concerning historical aerial images, there is a lack of overview of aerial image archives with regard to their spatial and temporal coverage. In addition, many archives of historical aerial images are not freely accessible and images collected for military purposes have yet to be declassified or have restricted access. One additional problem is the image quality, which is often compromised by photo processing, physical storage conditions, and the digitisation process, i.e., the distortion introduced during the scanning and the resolution of the scan (Stark et al., 2022). Furthermore, external information such as camera calibration certificates required for accurate photogrammetric processing, is often missing. Nevertheless, automated methods using SfM are currently being developed to process historical aerial images and generate time series DEMs and high-resolution orthoimages (Knuth et al., 2023). Images from the declassified stereo spy satellites from the Corona (Dehecq et al., 2020; Ghuffar et al., 2023) and Hexagon KH-9 missions can be easily accessed via the USGS Earth Explorer portal (free of charge for images that have already been scanned) and offer great potential for DEM reconstruction for geomorphic change detection (Maurer et al., 2015). However, despite their global coverage, their period of acquisition ranges between the 1960s and 1980s and their ground resolution from approximately 2 m to 8 m. From the year 2000, private companies often conduct airborne digital photogrammetric and Lidar surveys, but rarely make the data available. Modern high-resolution stereo satellite images with metric to sub-metric resolution (e.g. Worldview and*

*Pléiades) are commercially available after 2005 and can be used for large-scale DEM reconstruction, but they are not freely available and have limited temporal resolution. Consequently, studies in Alpine catchments with high spatial and temporal resolution datasets dating back to the last century are still very limited, preventing the comparison of spatio-temporal variation in sediment dynamics and the assessment of the response of catchment-scale sediment yield to climate change."*

*Stark, M., Rom J., Haas F., Piermattei L., Fleischer F., Altmann M., Becht M: Long-term assessment of terrain changes and calculation of erosion rates in an alpine catchment based on SfM-MVS processing of historical aerial images. How camera information and processing strategy affect quantitative analysis. Journal of Geomorphology: 43-77, 2022.*

*Maurer, J. and Rupper, S.: Tapping into the Hexagon spy imagery database: A new automated pipeline for geomorphic change detection. ISPRS Journal of Photogrammetry and Remote Sensing, 108, pp.113-127, 2015.*

*Ghuffar, S., Bolch, T., Rupnik, E. and Bhattacharya, A.: A Pipeline for Automated Processing of Declassified Corona KH-4 (1962–1972) Stereo Imagery. IEEE Transactions on Geoscience and Remote Sensing, 60, pp.1-14, 2022*

*Dehecq, A., Gardner, A.S., Alexandrov, O., McMichael, S., Hugonnet, R., Shean, D. and Marty, M.: Automated processing of declassified KH-9 Hexagon satellite images for global elevation change analysis since the 1970s. Frontiers in Earth Science, 8, p.566802, 2020.*

How likely is it to create directly comparable results for different selected study sites within the European Alps and worldwide? Please judge also the possible restrictions.

*--- Thank you for this question. We clarified the possibility of comparing results for different catchments in the discussion.*

*Comparing sediment balances between different study sites is very challenging due to the complexity of the mountain environment and the high dynamics of the river system in proglacial areas, which are subject to different forcing and with varying lengths of deglaciation periods. In addition, differences in the dataset and survey period may introduce another source of «lacking comparability» between the study areas. However, if data of comparable quality and spatial/temporal resolution are available, it is possible to identify site-specific developments and/or common trends, as well as general processes affecting the proglacial area and Alpine catchments.*

*"The emergence of new glacial forefields previously ice-covered is the first evidence of unprecedented rapid glacier melt caused by rising temperatures and longer heat waves that are affecting all Alpine glaciated catchments worldwide. Carrivick et al. (2013) and Baewert and Morche (2014) in their study show that erosion is the dominating process that takes place in the proximal area of the glacier and accumulation generally occurs in the distal area. Similarly, Anderson and Shean (2021) in their study of proglacial erosion rate found that exported materials tended to accumulate in large deposits below the proglacial limits, to then be distributed over subsequent decades or centuries. However, we show that fluvial sediment storage varies considerably depending on factors such as the local topography of the newly exposed active floodplain (bedrock versus sediment glacier bed, confinement versus presence of "accommodation space"), increasing floodplain area with the formation of new channel, sediment connectivity, and the subsequent sediment source from the lateral moraine. Thereby, retreating glaciers may uncover clean bedrock or may reveal large sediment sources stored in their former lateral and ground moraines. The amount of these glacigenic sediments depends on the sediment balance of the retreating glacier, which is a function of the sediment production of the rock wall erosion and the erosional potential of the glacier runoff, i.e. slope, balance, and precipitation (Zemp et al. 2005).*

*In addition, a single localized event such as an extreme precipitation event or the outburst of a subglacial water pocket can cause massive changes in the channel system, as also concluded by Anderson and Shean (2021), and numerous studies on the European Alps (e.g. Baewert and Morche, 2014, Lane et al. 2016, and Carrivick et al. 2013; 2018) and Himalaya (Cook et al., 2018). Furthermore, Buter et al. (2022) and Savi et al. (2023) note that intense rainfall events play a critical role in promoting and enhancing functional sediment connectivity (Wainwright et al., 2011) among landforms that are already structurally connected. In addition, extreme events can modify the structural connectivity by removing barriers to sediment fluxes (Turley et al. (2021)."*

---

## Author Comment (AC2)

*Dear Reviewers,*

*Thank you for your comments. We revised the manuscript considering all the reviewers' feedback.*
*We improved the abstract by highlighting the main findings of the study. As suggested, we clarified in the introduction the main hypotheses that guided the work. We have also substantially expanded the discussion section, including a discussion on the feasibility of generating similar datasets, a comparison with previous studies, as well as addressing all the points raised in the introduction such as sediment connectivity and 'peak water'. We also improved the figures based on reviewer feedback and updated the reference list.*
*Below are all the point-by-point responses to the reviewer's comments.*

**Response to the reviewer's comments**

**RC2: 'Comment on esurf-2022-63, Anonymous Referee #2**

In their manuscript, Piermattei et al. examine morphological changes of a proglacial river in the Kaunertal, Austria, using remotely sensed imagery that spans 66 years between 1953 and 2019. Using publicly available historical aerial imagery, they construct orthoimages and digital elevation models using structure from motion photogrammetry.

They combine these data sets with UAS derived orthoimages and DEMs as well as LiDAR derived ALS data to investigate volumetric changes of six river reaches over 19 epochs.

Exploiting this impressive data set, the authors gain interesting and valuable insight into the dynamics of proglacial areas. Also grounding on hydrological data, they find that river sediment loads are likely increasing in connection with elevated runoff contribution from melting Gepatschferner glacier. The authors further investigate the contribution from lateral sediment storage to the channel segments and shed light on the propagation of signals by subdividing the reaches into smaller segments.

The data and results presented in this very well written manuscript address a topic that is interesting for a broad range of readership and clearly merits publication in ESURF. Below, I have outlined some general points and a number of specific issues, all rather minor in nature that should be addressed before the manuscript can be accepted for publication.

*--- Thank you very much for the careful reading and very constructive review!*

General comments

As stated above, the manuscript is very well written, which is especially true for the excellent introduction. However, some of the aspects touched upon in the introduction, e.g. the "peak water" effect and how it will change the future behaviour of a proglacial river, or the aspect of connectivity, do not find their way into the discussion. It would be desirable to shed light on these aspects in the discussion. Furthermore, the relevance of the study for the sediment management of large reservoirs in the alps could be highlighted more pronouncedly in the manuscript, as the study contributes important insight into this direction.

*--- Thank you for this comment. We extended the discussion by adding a paragraph on the relevance of peak water and peak sediment in the Alpine catchment and their impact on the Alpine sediment dynamics, as well as sediment management for the reservoir. We also discuss more clearly the role of lateral moraines and sediment connectivity.*

*"In their recent review, Zhang et al. (2022) present a global inventory of increases in erosion and sediment yield resulting from cryosphere degradation. Their findings suggest that sediment transport will continue to increase until it reaches a maximum (sediment peak). This trend is linked to alterations in seasonal water supply, including increased winter liquid precipitation, early snowmelt and ice melt, and reduced snow-melt supply in later summer months, as also stated by Lane and Nienow (2019). Huss & Hock (2018) also noted that the "peak water", i.e. the maximum runoff from glacier long-term storage, will be reached in the coming decades due to ongoing glacier shrinking. All these factors will affect the hydrological and geomorphological processes of glaciated catchments with consequences for ecological functioning of Alpine streams, water-related hazards, downstream water availability, sediment management in large reservoirs, and hydropower production (Schaefli et al., 2019). We estimate an overall negative balance that should be taken into account in the management of sediment*

*deposits, as eroded sediments are likely to be mobilized downstream into the reservoir. However, the dynamics of sediment transport in Alpine river systems are complex and vary between catchments and landforms types (Savi et al., 2023). Therefore, each site has to be analysed separately with regard to sediment management strategies.*

*Overall, predicting sediment dynamics in a warming world is not yet well established and required further research on integrating sediment observations from multiple sources, developing sediment-transport models, and enhancing interdisciplinary and international scientific collaboration (Zhang et al., 2022)."*

**Specific Comments**

- L31-33: The authors might want to highlight their contribution to a better understanding of high-mountain sediment dynamics more precisely here.

*--- We agree, thank you. We added a few sentences summarizing the main finding of the study.*

*"The proglacial river in our study is in transition following massive glacier retreats (~1.8 km) creating new channel networks, leading to a progressively negative cumulative sediment balance for the entire study area. We found that high-magnitude meteorological and hydrological events associated with local glacier retreats have a strong impact on the sediment balance. According to the gauge record, there is an increase in such events as well as in runoff and probably in sediment transport capacity. Despite this, the last decade shows an overall decline in sediment supply that can be explained by a lower contribution of the lateral moraines coupled to the channel network, and less sediment sourced from the melting Gepatsch glacier as evidenced by the roches moutonnée exposed in the current/most recent forefield. However, considerable erosion (and thus delivery downstream) has been observed in the southern tributary."*

- L58: Hock et al., 2019

*--- Corrected*

- L95-109: I recommend rephrasing this section and focus this part of the introduction more clearly towards outlining the aims and underlying hypothesis of the study.

*--- Thank you for the suggestion. We rephrased this section focusing more on the purpose of our study, and formulating the main hypotheses that guided the work and related analyses.*

*"In this study, we analyse past and recent proglacial river changes and sediment storage of the main channel network of the Kaunertal Alpine catchment located in Austria with the overall aim of identifying links between channel changes, sediment availability/delivery, and hydro-meteorological forcing. We assume that river sediment loads are likely to increase due to increasing snow and glacier melt runoff, facilitated by climate changes. In addition, increasing frequencies of heavy precipitation could enhance hillslope-channel coupling. We further hypothesize that hillslopes that have experienced geomorphic changes in the past and are/were coupled to the channel network drive the most significant changes in the channel network. To validate these hypotheses, we reconstruct and quantify 66 years of sediment and river changes in the glacier forefield between 1953 and 2019 by applying the morphological method (Vericat et al., 2017) to DEMs from historical and digital aerial images and LiDAR that span inter-survey periods ranging from one month to 16 years (Fig. 1).... We also analysed the discharge data measured by the gauging station, focusing on discharge peaks and strong events, seasonal variation, and total trend. "*

- L97-98: This sentence seems misplaced here, consider moving towards the discussion or conclusion.

*--- Thank you. We removed the sentence. The concept is already stated in the conclusion and discussion section.*

- L112-113: repetition of L98-99, consider deleting here or in the introduction.

*--- Thank you. We removed it from the introduction, where we only mention the name of the catchment.*

- L118-119: "outlet of the Gepatsch reservoir"? I guess the authors want to refer to the outlet of the Fagge into the reservoir?

*--- Thank you, corrected.*

- L132-133: "Günther; Patzelt, Gernot (2015)"? This reference does not appear in the list of references.

*--- Thank you, the reference was not correct. We corrected in the text and added it to the reference list. Groß, G. and Patzelt, G.: The Austrian Glacier inventory for the Little Ice Age maximum (GI LIA) in ArcGIS (shapefile) format, 2015.*

- L147: Altmann et al., 2020

*--- Corrected*

- L161: The reference to Pfeifer et al. 2014 is also not included in the references.

*--- Added to the reference list*

- Table 1: The values for mean floodplain and mean channel slope seem very high here. Is there any chance that the unit is not degrees as given, but in percent? Furthermore, compared to the mean channel slope, the floodplain slope of some reaches is very high. I think the readers would appreciate details on how the floodplain slope was calculated here.

*--- Thank you for your question. The unit is correct and we have added a longitudinal profile in Figure 2 that shows the length of the river for each reach in relation to the elevation and the variation of the glacier over time. As can also be seen in the figure and Fig. 10, the proglacial area is very steep, with the exception of the two braided river reaches downstream (reach 1 and 2). In the case of the floodplain, we calculated the slope as the average of the digitized area. The main reason for the high slope values of the active floodplain is the inclusion of the steep lateral riverbank. However, to be more consistent with the longitudinal profile shown in Fig. 2, we recalculated the average slope from the delta Z (the elevation difference) between the in- and outflow of the respective reach and the distance along the thalweg. We clarified this in the text.*

[Figure]

- L217: Figure order?

*--- Thank you for the observation. However to minimize the number of figure we include the cross-section illustration in figure 5. If this is fine, we would not change this.*

- L224-226: The discrepancies between floodplain and channel slope would suggest that channel incision into the deposits is also a source of sediments in this setting.

*--- Yes exactly. Channel incision and erosion of the lateral riverbanks, which have been identified as an important factor of lateral connectivity by Cienciala et al. (2020), are evident by alterations in the floodplain area, particularly in the braided system, and contribute to sediment supply. We added a sentence about this in the results section.*

*Cienciala, P., Nelson, A.D., Haas, A.D. and Xu, Z.: Lateral geomorphic connectivity in a fluvial landscape system: Unraveling the role of confinement, biogeomorphic interactions, and glacial legacies. Geomorphology, 354, p.107036, 2020.*

- L277-278: You might want to either use "spatio-temporal" or "spatial-temporal"

*--- Thank you. We use "spatio-temporal" consistently throughout the document.*

- L296: Missing information seems to be rather highlighted as a hatched area than an oblique line?
*--- Thank you. We corrected in the figure caption.*

- L296-297: So, the unit of the color scale (m3 yr-1) does not apply to the epochs 2012-07, 2012-09, and 2012-10? I would think it would be good to somehow make this also clear in the figure itself.
*--- Yes exactly, as it does not make sense to derive m3 yr-1 for monthly estimates. Thank you for the suggestion. We added a * in the figure to mark these periods and updated the figure caption.*

[Figure]

- Figure 6b: If I understand the figure correctly, the size of the reaches changes over time? Would be interesting to see the sediment balance normalized to the respective reach area.
*--- Thank you for the interesting observation. Yes, that is correct. We calculated the normalized cumulative net sediment balance in meters, which corresponds to an area-weighted mean elevation change and we added a few sentences to the results. "Analysing the cumulative net sediment balance normalized by the area of the floodplain (i.e. the elevation change), a similar trend emerges, but the cumulative net balance of all connected reaches shows less variation. In addition, the negative contribution of Reach 1 is less dominant due to its large area. Reach 6 shows the highest amount of erosion with a cumulative elevation change of up to -5 m followed by reach 4. Interestingly, all reaches show a stable trend toward aggregation after 2012."*

[Figure]

- L321-323: Might there be a way to also show this graphically? Maybe use filled circles when the reach is included in the net balance, and open circles if not?

*--- Thank you for the good idea. We implemented your suggestion in the new figure (see above).*

- L351: There is no explanation of how the trend analysis was done. Is this based on an ordinary least squares regression?

*--- Thank you for the observation. We used the Mann-Kendall test to identify the presence of a trend in the maximum yearly runoff ($m^3/s$). The test provides the p-value as well as the slope and intercept that are used to derive the linear regression. We clarified this in the method section and we added in the method the reference of the python package used for the trend analysis.*

- L362-363: use Greek letter for sigma as before

*--- Thank you, corrected.*

- L364-365: As the authors pointed out in L265-266, volumetric mass loss might also be associated with melting of dead ice in the proglacial area, most likely in lateral moraines. This effect is certainly very difficult to quantify, but might be addressed in a short statement in the discussion.

*--- Thank you. Indeed, we used the DEM of differences to exclude dead ice in the proglacial area and lateral moraines as reported in L189, 190. We added a short sentence in the discussion.*

- L371-381: This is a very interesting and important analysis. Certainly, the difference between net volume changes with or without the river channels depends on the water area of the river reach, as you point out here. It would be interesting to see if and how Fig. 6a would change with such an analysis.

*--- Thank you for this observation and we agree that this is an important analysis. We clarified in the discussion the reason for including the water in the calculation of the sediment balance after showing that the area covered by water can reach more than 60% of the river reach.*
*"Therefore, excluding the area covered by water from the sediment balance calculation would also introduce an error, as we would exclude the change there. Baewert and Morche (2014) also mentioned this issue, concluding that in the case of a braided system, the shift of the river course between different surveys can compensate for this error to some extent, but this depends on the area covered by water. Bathymetric reconstruction of the riverbed from LiDAR and photogrammetric DEM is still a subject of research, and most studies of river sediment based on DEM of differences do not exclude the water area in the calculation (e.g. Calle et al., 2020; Scorpio et al., 2022; Savi et al., 2023). Only a few studies include the refraction by water during SfM photogrammetric reconstruction of the river surface (Dietrich, 2017; Lane et al., 2020). This development should be considered when UAV-based SfM photogrammetry is applied to derive erosion and deposition analysis of river environments from DEM, but it requires clear water conditions (Bertalan et al., 2023) that are rare in a glacier-fed stream. Moreover, it is unclear whether the bathymetric correction works as well for historical aerial photographs as for modern high-resolution UAV-based imagery."*

- L374-377: Might this be an error and the authors rather want to refer to Fig. 9 in this section?

*--- Done, thank you for spotting.*

- L377-381: This is certainly an interesting development, but it requires clear water conditions that are rare in a glacier-fed stream.

*--- Thank you for the comment, we added to the discussion (see paragraph above).*

- L390: You might want to add a reference to Fig. 6 here?

*--- Yes, agreed.*

- L408: superscript missing

*--- Done, thank you for spotting.*

- L413: impacted instead of "is driven by"?

*--- Corrected.*

- L413-414: Consider rephrasing, unclear what the increasing trend refers to.

*--- We agree, thank you. We rephrased the sentence.*

- L420: In my view, it would be good to refer to the river reaches as "R1" to "R6". But this is the only instance in the manuscript where this is done. Consider harmonizing.

*--- Thank you for spotting this. We prefer to reduce the use of acronyms in the text so we removed R1 and R6 and we report the number only.*

- L425-430: In this section a thorough comparison between the data presented here and the work of Baewert and Morche (2014) would be interesting. Here the authors find aggradation in all (but one) river reaches. Baewert and Morche (2014), however, find widespread aggradation following this August 2012 event. While this is certainly related to the different data sets used that result in different survey periods and areas, a comparison is interesting, as it also underlines the dynamics of the proglacial area in this setting.

*--- Thank you for the suggestion. We added a more detailed comparison with the study of Baewert and Morche (2014). As also suggested, we can assume that the discrepancies in terms of sediment volume and overall trend are related to different datasets in terms of survey period and river reach delineation. "In contrast, in the work of Baewert and Morche (2014) in the Fagge River, aggradation dominated their five river reaches but one proximal to the glacier with a volume change of about -69000 m3 between June 2012 and September 2012 with an erosion of up to 5 m in the bed channel. While the reason for this discrepancy is certainly related to the different datasets used, which resulted in different survey periods and areas - they worked on disconnected smaller river reaches, while we derived the balance from all connected river reaches - this comparison also underlines the dynamics of the proglacial area in this setting."*

- L430-439: Certainly, the work of Anderson and Shean (2021) is very close to the presented study and deserves attribution here. But there are numerous studies from the European Alps that also deserve a reference here, e.g. Lane et al. 2016 (and other works from Stuart Lane's group), and the works of Carrivick et al., or Baewert and Morche already cited at other places in the manuscript.

*--- We agree, thank you. We added these references in the text.*

- L437: Can the bedrock be seen in the orthophotos, or is this additional field evidence?

*--- We do have field evidence, however, it can also be seen in the orthophoto in figure 10c and the photo taken in the field at the time of the UAV acquisition. We added the reference to the figure in the discussion.*

- L480: use m3 instead of cubic meters here?

*--- Done.*

- L490: Either use L.P. or LP here.

*--- Corrected with LP.*